# Molecular classification and tumor microenvironment characteristics in pheochromocytomas

Sen Qin[1†], Yawei Xu[1†], Shimiao Yu[1], Wencong Han[1], Shiheng Fan[2], Wenxiang Ai[2], Kenan Zhang[1], Yizhou Wang[1], Xuehong Zhou[1], Qi Shen[1], Kan Gong[1*], Luyang Sun[1*], Zheng Zhang[1*]

[1]Department of Biochemistry and Molecular Biology, School of Basic Medical Sciences, Department of Urology, Peking University First Hospital, Peking University Health Science Center, Beijing, China; [2]Shenzhen Institute of Ladder for Cancer Research, Shenzhen, China

*For correspondence:
gongkan_pku@126.com (KG);
luyang_sun@hsc.pku.edu.cn (LS);
doczhz@aliyun.com (ZZ)

[†]These authors contributed equally to this work

Competing interest: The authors declare that no competing interests exist.

**Abstract** Pheochromocytomas (PCCs) are rare neuroendocrine tumors that originate from chromaffin cells in the adrenal gland. However, the cellular molecular characteristics and immune microenvironment of PCCs are incompletely understood. Here, we performed single-cell RNA sequencing (scRNA-seq) on 16 tissues from 4 sporadic unclassified PCC patients and 1 hereditary PCC patient with Von Hippel-Lindau (VHL) syndrome. We found that intra-tumoral heterogeneity was less extensive than the inter-individual heterogeneity of PCCs. Further, the unclassified PCC patients were divided into two types, metabolism-type (marked by NDUFA4L2 and COX4I2) and kinase-type (marked by RET and PNMT), validated by immunohistochemical staining. Trajectory analysis of tumor evolution revealed that metabolism-type PCC cells display phenotype of consistently active metabolism and increased metastasis potential, while kinase-type PCC cells showed decreased epinephrine synthesis and neuron-like phenotypes. Cell-cell communication analysis showed activation of the annexin pathway and a strong inflammation reaction in metabolism-type PCCs and activation of FGF signaling in the kinase-type PCC. Although multispectral immunofluorescence staining showed a lack of CD8+ T cell infiltration in both metabolism-type and kinase-type PCCs, only the kinase-type PCC exhibited downregulation of *HLA-I* molecules that possibly regulated by *RET*, suggesting the potential of combined therapy with kinase inhibitors and immunotherapy for kinase-type PCCs; in contrast, the application of immunotherapy to metabolism-type PCCs (with antigen presentation ability) is likely unsuitable. Our study presents a single-cell transcriptomics-based molecular classification and microenvironment characterization of PCCs, providing clues for potential therapeutic strategies to treat PCCs.

## eLife assessment

This **valuable** study advances our understanding of the potential therapeutic strategies for the treatment of pheochromocytomas using single-cell transcriptomics. The authors propose a new molecular classification criterion based on the characterization of tumor microenvironmental features, based on **solid** evidence. The work, which could be improved further through delineating the choice of the PASS scoring system, will be of broad interest to clinicians, medical researchers, and scientists working in the field of pheochromocytoma.

## Introduction

Pheochromocytomas (PCCs) are rare neuroendocrine tumors that originate from chromaffin cells in the adrenal gland (*Fishbein et al., 2017*; *Liu et al., 2018*; *Nölting et al., 2022*). PCCs have been shown to display a remarkable diversity of driver alterations, including germline and somatic mutations as well as somatic fusion genes (*Favier et al., 2015*; *Pillai et al., 2017*; *Toledo et al., 2017*). This diversity is reflected in the current molecular taxonomy of PCCs The Cancer Genome Atlas (TCGA), including for example the pseudo-hypoxic type (germline mutations in *SDHx*, *FH*, and *VHL*, etc.), the kinase-signaling type (germline or somatic mutations in *RET*, *NF1*, *TMEM127*, *MAX*, and *HRAS*, etc.), and the Wnt-signaling type (somatic mutations in *CSDE1* and somatic gene fusions affecting *MAML3*; *Crona et al., 2017*). However, 50~60% of PCCs fail to be classified using this system (*Lenders et al., 2014*).

As understanding of PCC clinicopathological and immunophenotypic characteristics has deepened, the former concept of 'benign or malignant PCC' has been abandoned; since 2017, all PCCs are regarded as malignant tumors with metastatic potential according to the WHO pathological classification (*Lam, 2017*; *Mete et al., 2022*). There is no single histo-morphological feature indicating the metastasis risk of PCCs; a number of multifactorial systems have been proposed (*Kulkarni et al., 2016*). A cytomorphometric study reported that extra-adrenal location, coarse nodularity, confluent necrosis, and the absence of hyaline globules are associated with metastasis risk (*Lewis, 1971*). Several scoring systems that for example consider vascular invasion, tumor size, diffuse growth, and mitotic activity have been used for risk assessment of metastasis risk of PCCs (*Sherwin, 1959*; *Sisson et al., 1984*; *Symington and Goodall, 1953*). A scoring system—the Pheochromocytoma of the Adrenal gland Scaled Score (PASS)—is weighted for histologic features and has been used to separate tumors with a potential for a biologically aggressive behavior (PASS > or = 4) from tumors with benign lesion (PASS < 4) (*Thompson, 2002*).

Immune checkpoint inhibitors (ICIs) have achieved remarkable results in a variety of solid tumors in recent years (*Liu et al., 2022*; *Zhang et al., 2020*). Currently, there are only two clinical trials investigating the efficacy of ICIs for the treatment of metastatic PCCs: one is still in the recruitment stage, and the other showed 43% non-progression rate (NPR) and 0% overall response rate (ORR) in seven PCC patients (*Dai et al., 2020*; *Jimenez et al., 2022*; *Naing et al., 2020*). The composition of immune cells in the tumor immune microenvironment is an essential indicator for predicting likely responses to immunotherapy, and serves as the basis of immunotherapy (*Liu et al., 2023*; *Liu et al., 2022*). Despite the application of immunohistochemical staining to characterize the PCC microenvironment (*Calsina et al., 2023*; *Tufton et al., 2022*), limited knowledge exists regarding the cell composition and intercellular crosstalk within the PCC microenvironment.

In the present study, we performed single-cell RNA sequencing (scRNA-seq) analysis of 11 tumor tissues and 5 adjacent normal adrenal medullary tissues from 4 sporadic PCC patients with unclassified mutations and 1 hereditary PCC patient with Von Hippel-Lindau (VHL) syndrome caused by germline mutation in *VHL*. We found less intra-tumoral heterogeneity than inter-individual heterogeneity of PCCs. For the unclassified PCCs, we distinguished metabolism-type and kinase-type PCCs based on copy number variants (CNVs) and gene expression profile data. These two PCC types also displayed distinct characteristics of tumor evolution and cell-cell communication. Although multispectral immunofluorescence staining showed a lack of CD8+ T cell infiltration in both metabolism-type and kinase-type PCCs, only the kinase-type PCC exhibited downregulation of *HLA-I* molecules that possibly regulated by *RET*, suggesting the potential of combined therapy with kinase inhibitors and immunotherapy for kinase-type PCCs, whereas the application of immunotherapy to metabolism-type PCCs (with antigen presentation ability) is likely unsuitable. Our study presents a single-cell transcriptomics-based molecular classification and microenvironment characterization of PCCs and provides clues for potential therapeutic strategies to treat PCCs.

## Results

### A landscape view of cell composition in PCCs

To characterize the molecular correlates of unclassified PCCs based on known prominent driver alterations, we performed single-cell RNA sequencing (scRNA-seq) on 16 collected specimens from 5 PCC patients (P1-P5). Four of these patients (P1-P4) suffered from sporadic PCCs: whole exome

sequencing (WES) of these tumor tissues detected somatic mutations as compared with their adjacent normal adrenal medullary tissues, including in *JAK2*, *ARHGEF39*, *KMT2D*, and *MST1* (**Supplementary file 1a**); note that none of these fit into the three previously molecularly defined groups in the TCGA molecular taxonomy (**Crona et al., 2017**). The cohort also included 1 hereditary PCC patient (P5) with Type 2 Von Hippel-Lindau (VHL) syndrome, with a germline missense mutation in *VHL* (**Supplementary file 1a**).

We collected one resected tumor specimen from P1 (P1_T1), three specimens from distinct intra-tumoral sites from P2 (P2_T1, P2_T2, and P2_T3) and P5 (P5_T1, P5_T2, and P5_T3), as well as two specimens from distinct intra-tumoral sites from P3 (P3_T1 and P3_T2) and P4 (P4_T1 and P4_T2). To enable comparisons, one normal adrenal medullary tissue adjacent to the tumor was collected from each patient (P1_A, P2_A, P3_A, P4_A, and P5_A) (**Figure 1A**). The morphology of PCC cells was assessed through hematoxylin-eosin-stained tumor tissues, showing the typical cell arrangement (small alveoli surrounded by fibro-vascular stroma) and characteristic shapes (polygonal or fusiform cells) of PCCs (**Lupşan et al., 2016**; **Figure 1—figure supplement 1**). Meanwhile, immunocytochemistry staining showed robust expression of Chromogranin A (CGA) in these tumor tissues (**Figure 1—figure supplement 1**). These pathological evaluations validate the collected samples as PCC tumor specimens (**Mete et al., 2022**). Additionally, we utilized the PASS scoring system to assess the histological features and evaluate the metastasis risk of PCC patients (**Supplementary file 1b**).

The specimens were digested into single cell suspensions, and 3'-scRNA-seq (Chromium Single Cell 3′ v3 Libraries) analysis was performed on each sample. After quality control filtering to remove cells with low gene detection and high mitochondrial gene coverage, we retained 133,894 individual cells (**Figure 1—figure supplement 2A–2C**). SCTransform normalization and principal component analysis (PCA) were then employed for unsupervised dimensionality reduction prior to clustering. Uniform Manifold Approximation and Projection (UMAP) was used for visualizing the distinct specimens, patients, and tissue types (**Figure 1B and C**).

We classified the 67 detected clusters into 13 cell types according to their expression profiles for recognized marker genes and Spearman correlation analysis between clusters: the classified cell types included adrenal cells (marked by *DLK1* and *RBP1*), endothelial cells (marked by *PECAM1* and *VWF*), fibroblasts (marked by *THY1* and *PLAC9*), smooth muscle cells (marked by *ACTA2* and *TAGLN*), monocytes/macrophages (marked by *CD14* and *CD163*), neutrophils (marked by *S100A8* and *S100A9*), T cells (marked by *IL7R* and *CD3D*), natural killer cells (NKs, marked by *KLRD1* and *GNLY*), proliferating cells (marked by *TOP2A* and *MKI67*), plasma cells (marked by *XBP1* and *IGKC*), B cells (marked by *MS4A1* and *CD79A*), mast cells (marked by *KIT* and *TPSB2*), and sustentacular cells (marked by *S100B* and *CRYAB*) (**Figure 1C and D**, **Figure 1—figure supplement 2D** and **Figure 1—figure supplement 3**). Thus, our clustering and cell type annotation analysis identified diverse adrenal cells, stromal cells, and immune cells (including lymphocytes and myeloid cells) within the PCC microenvironment (**Figure 1—figure supplement 4**).

## The intra-tumoral heterogeneity was less extensive than the inter-individual heterogeneity of PCCs

In light of the remarkable diversity of drivers including germline and somatic mutations reported among PCCs (**Crona et al., 2017**), we investigated PCC heterogeneity at the inter-individual and intra-tumoral levels based on the PASS system and single-cell transcriptional profiles. The PASS scores of 11 collected PCC tissues ranged from 2 to 9, showing obvious heterogeneity among the examined intra-tumoral sites and inter-individual comparisons for the 5 PCC patients (**Supplementary file 1b**). The Spearman correlation analysis based on average gene expression in each specimen showed that the intra-tumoral heterogeneity was less than inter-individual heterogeneity (**Figure 2A**), inconsistent with the PASS score evaluation. We further analyzed the cell type composition in each specimen and found similar fractions of cell types between the collected intra-tumoral sites despite different PASS scores (**Figure 2B**), indicating relatively small intra-tumoral differences in both gene expression and cell type composition. These results revealed that scRNA-seq analysis and PASS score represent different levels and dimensions of PCC heterogeneity.

The inter-individual correlation analysis indicated that P4 showed low correlation with the other three sporadic PCCs patients (P1-P3) (**Figure 2C**), and further analysis of cell type fractions showed a significantly elevated proportion of adrenal cells (77%) in P4; the adrenal cells comprised between 16 and

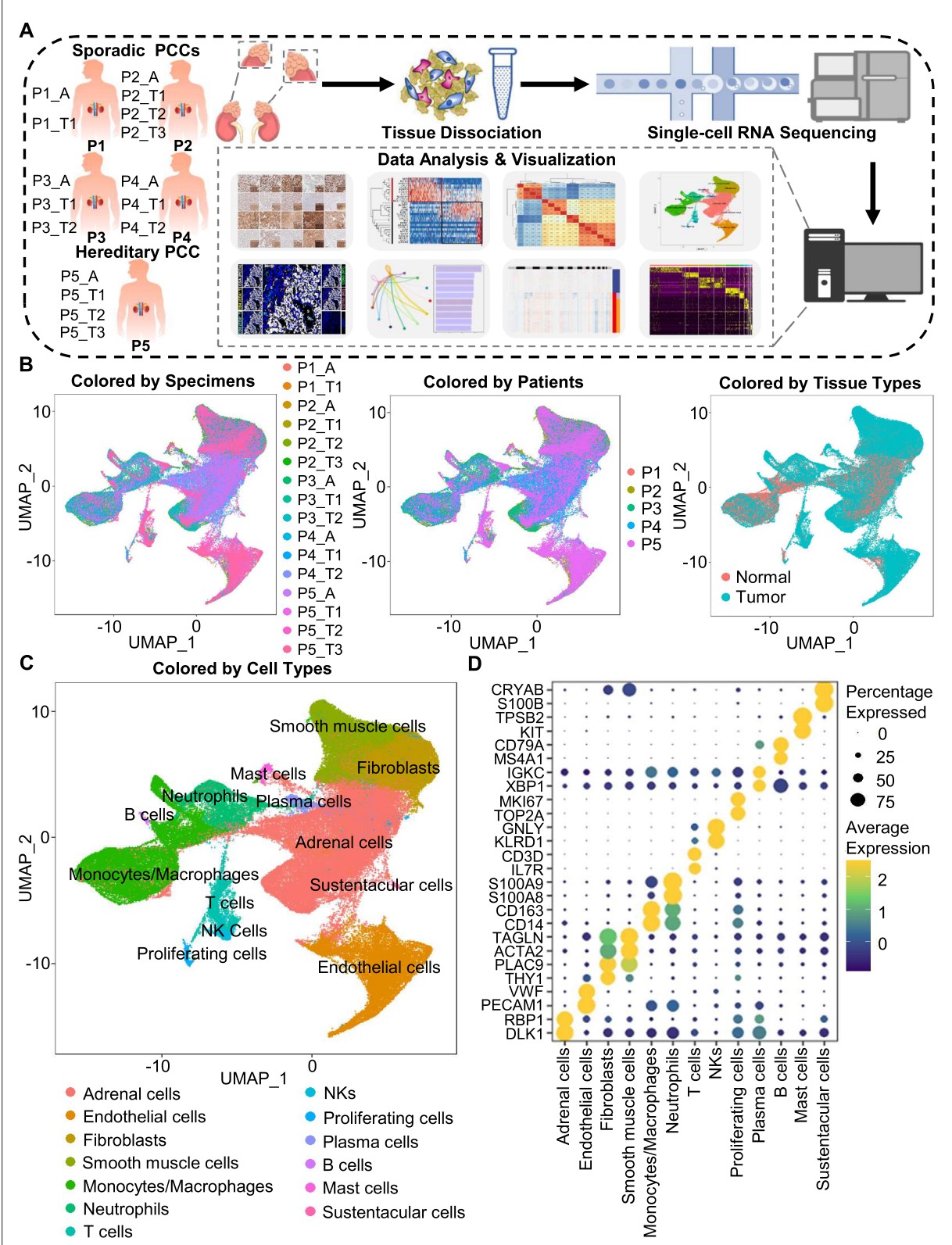

**Figure 1.** Integration analysis across 5 PCC patients revealing the cell composition of the PCC microenvironment. (**A**) Schematic of the experimental pipeline. Eleven tumor specimens and 5 adjacent normal adrenal medullary specimens were isolated from 5 PCC patients, dissociated into single-cell suspensions, and analyzed using 10 x Genomics Chromium droplet scRNA-seq. (**B**) UMAP plots illustrating 133,894 cells from 16 specimens across 5 PCC patients. Cells color-coded by specimens, patients, and tissue types. (**C**) UMAP plot showing 13 main cell types from all specimens. (**D**) Dot plot of

*Figure 1 continued on next page*

*Figure 1 continued*

representative marker genes for each cell type. The color scale represents the average marker gene expression level; dot size represents the percentage of cells expressing a given marker gene.

The online version of this article includes the following figure supplement(s) for figure 1:

**Figure supplement 1.** Hematoxylin-eosin staining and immunohistochemistry staining of CGA marker in formalin-fixed paraffin-embedded PCC tissue sections matched to scRNA-seq specimens.

**Figure supplement 2.** Quality control and cell clustering of scRNA-seq data.

**Figure supplement 3.** Correlation coefficient among cell clusters.

**Figure supplement 4.** Integration analysis across five PCC patients revealing the cell type composition of the PCC microenvironment.

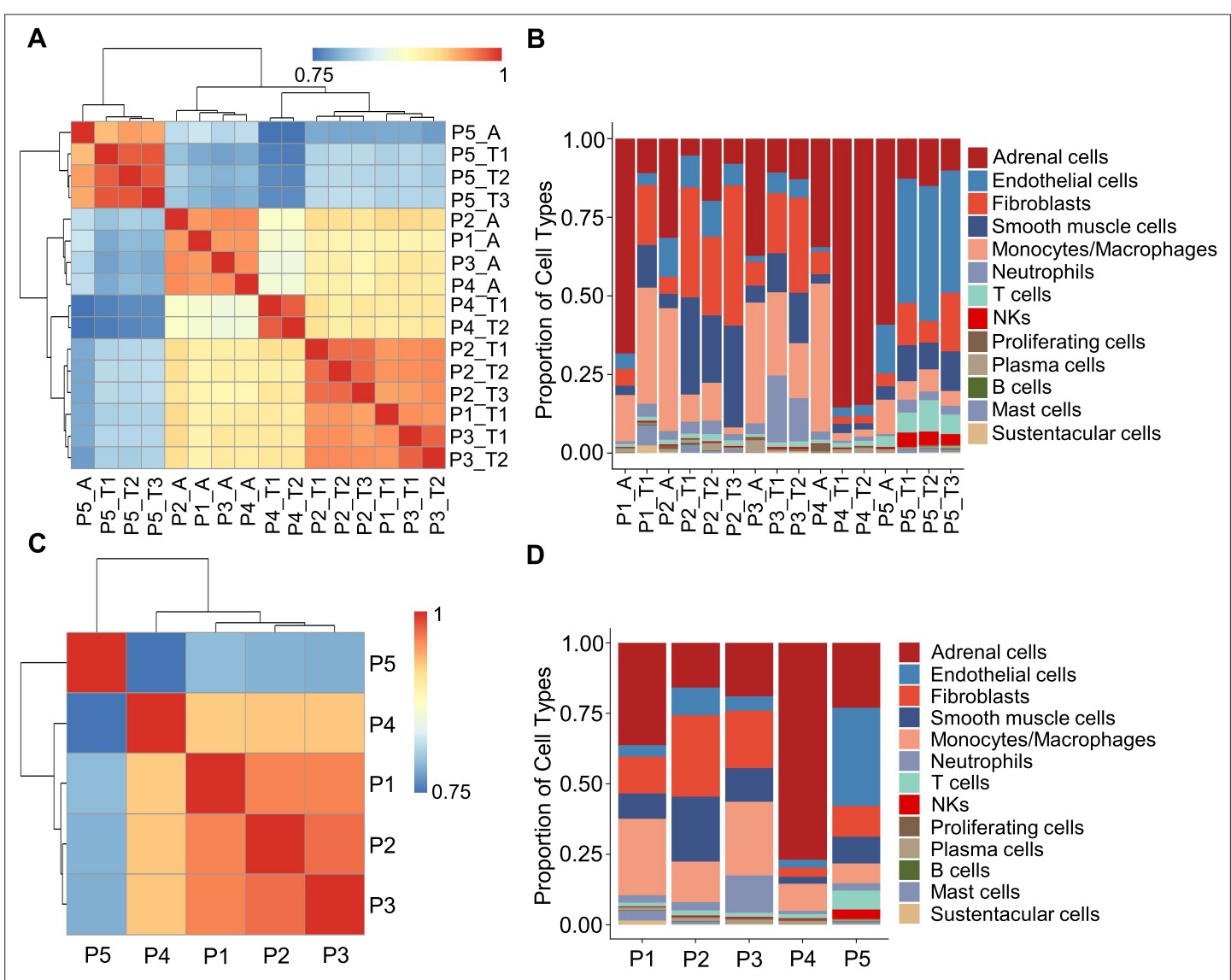

**Figure 2.** Correlation analysis reveals less intra-tumoral heterogeneity than inter-individual heterogeneity. (**A**) Heatmap plotting the correlation coefficient among 16 specimens. The color keys from blue to red indicate the correlation coefficient from low to high. (**B**) Bar plot showing the percentage of cell types in 16 specimens. (**C**) Heatmap of the correlation coefficient among five PCC patients. The color keys from blue to red indicate the correlation coefficient from low to high. (**D**) Bar plot depicting the frequency distribution of cell types in five PCC patients.

36% of the total cells in the P1-P3 specimens (*Figure 2D*). P4 also had an obviously distinct distribution of stromal cells *vs* immune cells (9% stromal cells and 14% immune cells) as compared to P1-P3 (28–62% stromal cells and 22–43% immune cells; *Figure 2D*). Consistent with the reported vascular endothelial hyperplasia phenotype in VHL syndrome (*Vortmeyer et al., 2013*), it was unsurprising that we observed poor correlation between the sporadic patients (P1-P4) and the VHL patient (P5), reflecting a higher proportion of endothelial cells in P5 (35%) as compared to P1-P4 (3–10%) (*Figure 2C and D*).

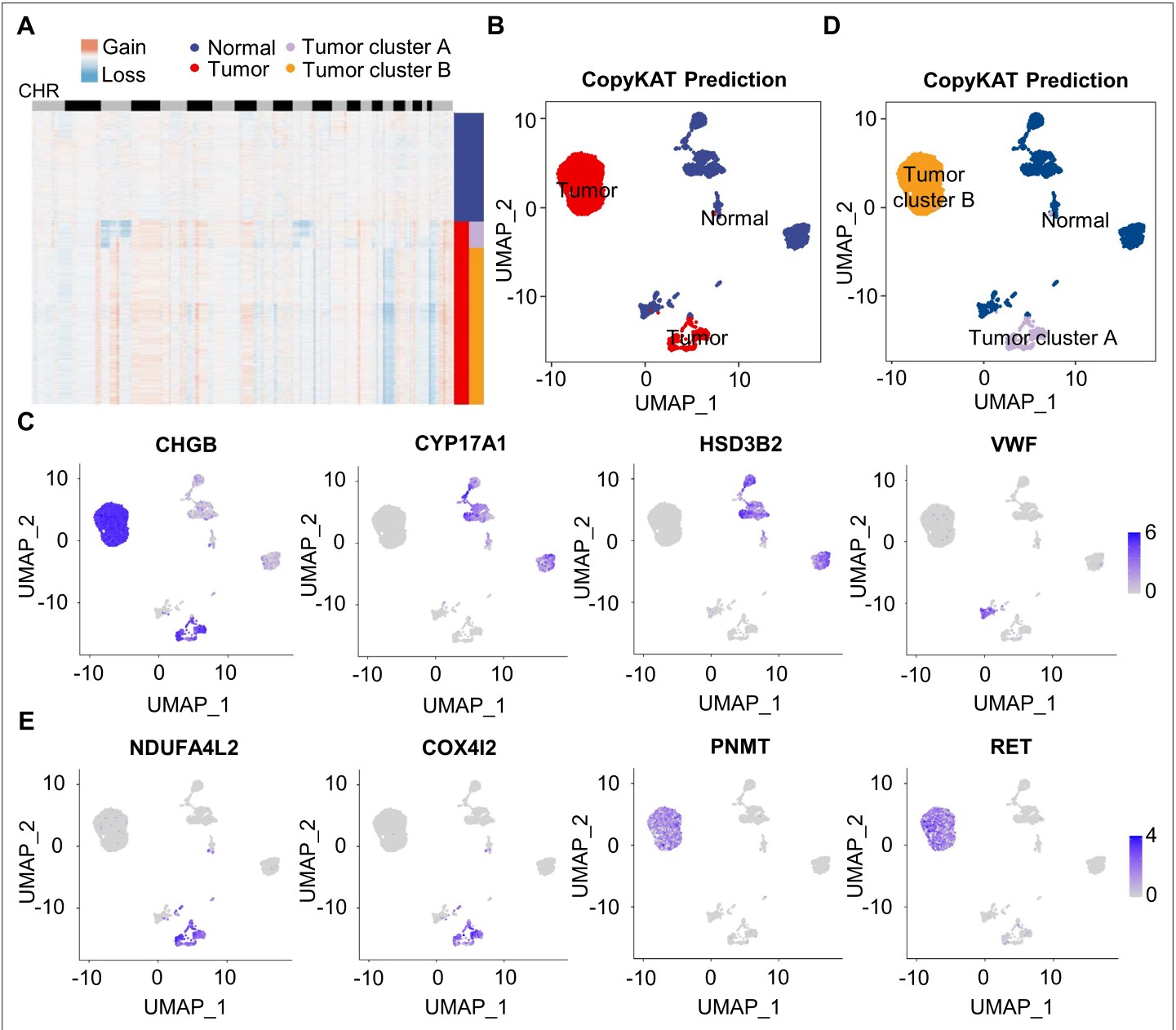

**Figure 3.** Single-cell copy number profiles in PCC clusters inferred by CopyKAT. (**A**) Heatmap indicating the CNV patterns of inferred normal cells, tumor cells, tumor cluster A, and tumor cluster B. Blue, white, and red respectively indicate deletion from a chromosome, normal chromosome, and amplification on a chromosome. (**B**) UMAP plot of the inferred normal cells (blue) and tumor cells (red) identified by CopyKAT. (**C**) Feature plots showing the marker gene expression levels in inferred normal cells and tumor cells. (**D**) UMAP plot depicting the inferred normal cells (blue), tumor cluster A (purple), and tumor cluster B (orange) identified by CopyKAT. (**E**) Feature plots displaying the marker gene expression levels in inferred tumor cluster A and B.

## Identification of tumor clusters through CNV analysis

To further characterize PCC cells, we collected adrenal cells, which theoretically include adrenocortical cells and chromaffin cells (*Yates et al., 2013*). The copy number karyotyping of aneuploid tumors (CopyKAT) algorithm (*Gao et al., 2021*) was applied, which estimates genomic copy number variants (CNVs) from scRNA-seq data by employing an integrative Bayesian segmentation approach to distinguish PCC cells from normal chromaffin cells. Our analysis identified two clusters among adrenal cells (normal cells and tumor cells), with tumor cells showing extensive chromosomal amplification and/or deletion (*Figure 3A*). Consistently, the inferred aneuploid tumor cells highly expressed *Chromogranin B* (*CHGB*), which is a marker of PCC cells (*Wiedenmann et al., 1988*; *Winkler and Fischer-Colbrie, 1992*; *Figure 3B and C*). The inferred diploid normal cells expressing high levels of *CYP17A1* and *HSD3B2* were defined as adrenocortical cells (*Kubota-Nakayama et al., 2016*), while the inferred diploid normal cells showing low *CHGB* expression were defined as normal chromaffin cells (*Figure 3B and C*). We also found a cluster of inferred diploid normal cells with high expression of *VWF* (a marker for vascular endothelial cells) in the VHL patient (P5) (*Figure 3B and C*).

We noticed two CNV patterns among the inferred aneuploid tumor cells, with the distinction evident in chromosomes 3, 11, and 17 (*Figure 3A*). These CNV profiles were reflected in distinct tumor clusters (tumor cluster A and B) through CopyKAT prediction (*Figure 3D*). We observed that cells in inferred tumor cluster A exhibited high expression of *NDUFA4L2* and *COX4I2*, while cells in inferred tumor cluster B expressed high levels of *PNMT* and *RET* (*Figure 3D and E*), suggesting the different gene expression between tumor cluster A and B. PNMT is known as an enzyme that catalyzes the conversion of norepinephrine to epinephrine and that RET is a member of the receptor tyrosine kinase family, while NDUFA4L2 and COX4I2 respectively function in the oxidative phosphorylation and glycolysis pathways (*Drilon et al., 2018*; *Mahmoodi et al., 2020*; *Sinkler et al., 2017*). Taken together, we distinguished PCC cells from adrenal cells through CNV analysis and found two clusters of PCC cells with different single-cell copy number profiles.

## Definition of metabolism-type and kinase-type tumors among unclassified PCCs

To explore the gene expression profile between tumor cluster A and B, we then used the FindAll-Markers function of the Seurat suite (*Butler et al., 2018*; *Satija et al., 2015*; *Stuart et al., 2019*) to identify differentially expressed genes (DEGs) of these tumor clusters (*Figure 4A and B*). A functional enrichment analysis based on gene ontology (GO) annotation was performed to further characterize the biological function of tumor clusters: the tumor cluster A DEGs showed functional enrichment for terms including generation of metabolites and energy, response to oxygen levels, response to hypoxia, and ATP metabolic process and some notably upregulated genes of tumor cluster A included known energy metabolism-related genes such as *NDUFA4L2*, *COX4I2*, *RGS4*, and *AQP1* (*Figure 4C*). For tumor cluster B, the GO analysis showed enrichment for terms including cytoplasmic translation, neuron projection development, axon development, and cell growth, and there was obvious upregulation of *PNMT*, *RET*, and genes encoding other kinases known to function in proliferation, as well as genes that participate in the regulation of neuroendocrine functions (e.g. *CALM1*, *PENK*, and *NPY*; *Figure 4D*).

Notably, immunocytochemistry staining of serial tumor sections showed strong accumulation of PNMT and RET in P4, while NDUFA4L2 and COX4I2 accumulation was characteristic for the P1-P3 and P5 sections, providing protein-level evidence to validate the differential trends detected from the scRNA-seq results (*Figure 4E*). As a consequence, we propose a metabolism-type (P1-P3) and kinase-type (P4) classification for unclassified sporadic PCCs. Consistently, kinase-signaling PCCs by TCGA classification include germline and somatic mutations in *RET*, and are uniquely able to secrete epinephrine, owing to their high expression of *PNMT*; P5 could also be classified into the metabolism-type we defined, which is known to display reprogramming of cellular energy metabolism caused by *VHL* mutation and subsequent *HIF* dysregulation (*Chappell et al., 2019*). When considering clinical information from five PCC patients, including tumor size, signs, symptoms and laboratory tests, we observed that P4 exhibited relatively higher blood pressures and the plasma levels of 3-methoxytyramine and normetanephrine compared to P1-P3 and P5 (*Supplementary file 1c*), consistent with the high expression of PNMT in the kinase-type PCC patient (*Figure 4E*). These results support that the previously unclassified PCCs can be classified as metabolism-type or kinase-type

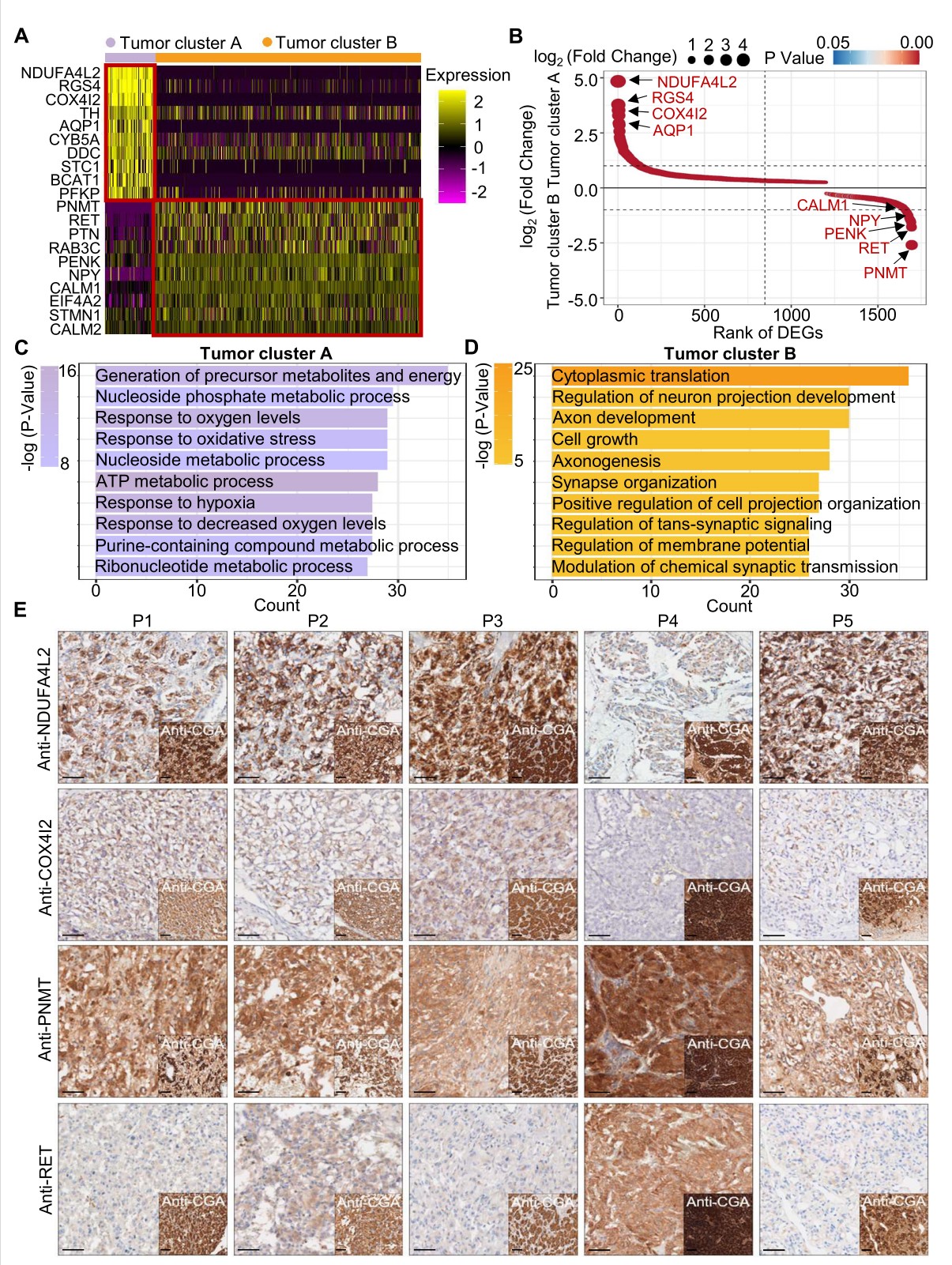

**Figure 4.** PCC patients were classified into metabolism-type and kinase-type. (**A**) Heatmap plotting the scaled expression patterns of major marker genes in each tumor cluster. The color keys from pink to yellow indicate relative expression levels from low to high. (**B**) Dot plot depicting up-regulated genes of tumor cluster A (top) and tumor cluster B (bottom). The x-axis specifies the rank of DEGs and the y-axis specifies the natural logarithm of the FC. Dotted vertical and horizontal lines reflect the filtering criteria. Dot size represents the natural logarithm of the FC of genes. The color keys from

*Figure 4 continued on next page*

*Figure 4 continued*

blue to red indicate the P-value from high to low. (**C, D**) GO enrichment analysis of the up-regulated genes in tumor cluster A (**C**) and tumor cluster B (**D**) indicating the top altered 10 terms in the biological process of gene ontology. The x-axis specifies the number of genes enriched in the pathways. The color keys from shallow to deep indicate the p-value from high to low. (**E**) Immunohistochemistry staining of CGA, NDUFA4L2, COX4I2, PNMT, and RET markers in formalin-fixed paraffin-embedded PCC tissue sections matched to scRNA-seq specimens. Scale bar, 100 μm.

based on transcriptional programs, with these types partially corresponding to the pseudo-hypoxic type and kinase-signaling type of PCCs from the TCGA classification.

## Trajectory analysis reveals subclonal dynamics of PCC types

We next examined tumor subclonal evolution patterns for the metabolism-type and kinase-type PCCs, by applying the Dynverse algorithm to order cells along a pseudotime trajectory (*Saelens et al., 2019*). We first clustered metabolism-type PCC cells into two subclusters based on DEGs and found similar proportions of tumor subcluster 1 and 2 at both the early stage and late stage of tumor evolution (*Figure 5A and B*). Late up-regulated genes include *S100A4* and *TAGLN* that promote epithelial-mesenchymal transition and cell invasion, which highly expressed in subcluster 2 (*Figure 5C*), suggesting the higher metastasis potential of subcluster 2 as compared with subcluster 1. Metabolism-related genes such as *NDUFA4L2*, *ATP5MG*, *NDUFB4*, and *COX17* were highly expressed at early stage, while *SOD3* and *COX7A1* were up-regulated at the late stage (*Figure 5C*), revealing the consistently active metabolism phenotype of metabolism-type PCCs cells. We also found the high expression of tumor suppressor gene *SPINT2* at early stage and subsequent up-regulation of the oncogene *JUNB* (*Figure 5C*), revealing an increase of metastasis potential over the tumor evolution of metabolism-type PCCs. In short, these results indicate that cells from metabolism-type PCCs display phenotype of consistently active metabolism and increasingly metastasis potential.

We then clustered kinase-type PCC cells into two subclusters and found that the major clone has been changed from subcluster 1–2 along the trajectory (*Figure 5D and E*). Further analysis of DEGs in the pseudotime trajectory showed early high expression of *CALM1* and *CALM2* in the subcluster 1, and late up-regulation of *RET* in the subcluster 2 (*Figure 5F*), suggesting the persistent dysregulation of kinase signals over tumor evolution. We also noticed the early expression of *PNMT*, while neuron-specific genes such as *MAP2*, *MAP1B*, *L1CAM*, and *CNTN1* were up-regulated later in the pseudo-time trajectory (*Figure 5F*), suggesting the decrease of epinephrine synthesis and the appearance of neuron-like phenotypes during tumor evolution, consistent with previously reported neuron-like phenotypes in PCC cells with high expression of RET (*Califano et al., 1995*; *Powers et al., 2003*; *Powers et al., 2009*). Taken together, these analyses have revealed the transcriptional clonal dynamics of metabolic-type and kinase-type PCCs.

## Cell-cell communication analysis to further characterize the tumor microenvironment

Having characterized differences in the cell type composition, gene expression, and clonal evolution between metabolism-type and kinase-type PCCs, we subsequently expanded our investigation to cell-cell communication occurring within the distinct tumor microenvironments of these two types of PCCs. We adopted CellChat tool to quantitively analyze inter-cellular communication networks (*Jin et al., 2021*). We observed the activation of the annexin signaling pathway in metabolism-type PCCs (*Figure 6A*). Further analysis of participated ligand-receptors showed major participation of *ANXA1-FPR1* pair in the annexin signaling pathway (*Figure 6B*), and high co-expression of *ANXA1* and *FPR1* in neutrophils and monocytes/macrophages (*Figure 6C*). It has been reported that the annexin signaling pathway is known to be activated under a strong inflammatory reaction (*Gastardelo et al., 2014*; *Gavins et al., 2003*; *Gerke and Moss, 2002*; *Hayhoe et al., 2006*). Combined with our findings of a higher proportion of neutrophils and monocytes/macrophages in metabolism-type as compared with kinase-type (*Figure 6—figure supplement 1*), we speculate that metabolism-type PCCs may display elevated inflammatory responses. We then performed Gene Set Enrichment Analysis (GSEA) on metabolism-type and kinase-type PCCs by using hallmark gene sets in the MSigDB databases (*Liberzon et al., 2011*). As expected, the inflammatory response signaling pathway showed enrichment in metabolism-type PCCs as compared with that in kinase-type PCC (*Figure 6D*).

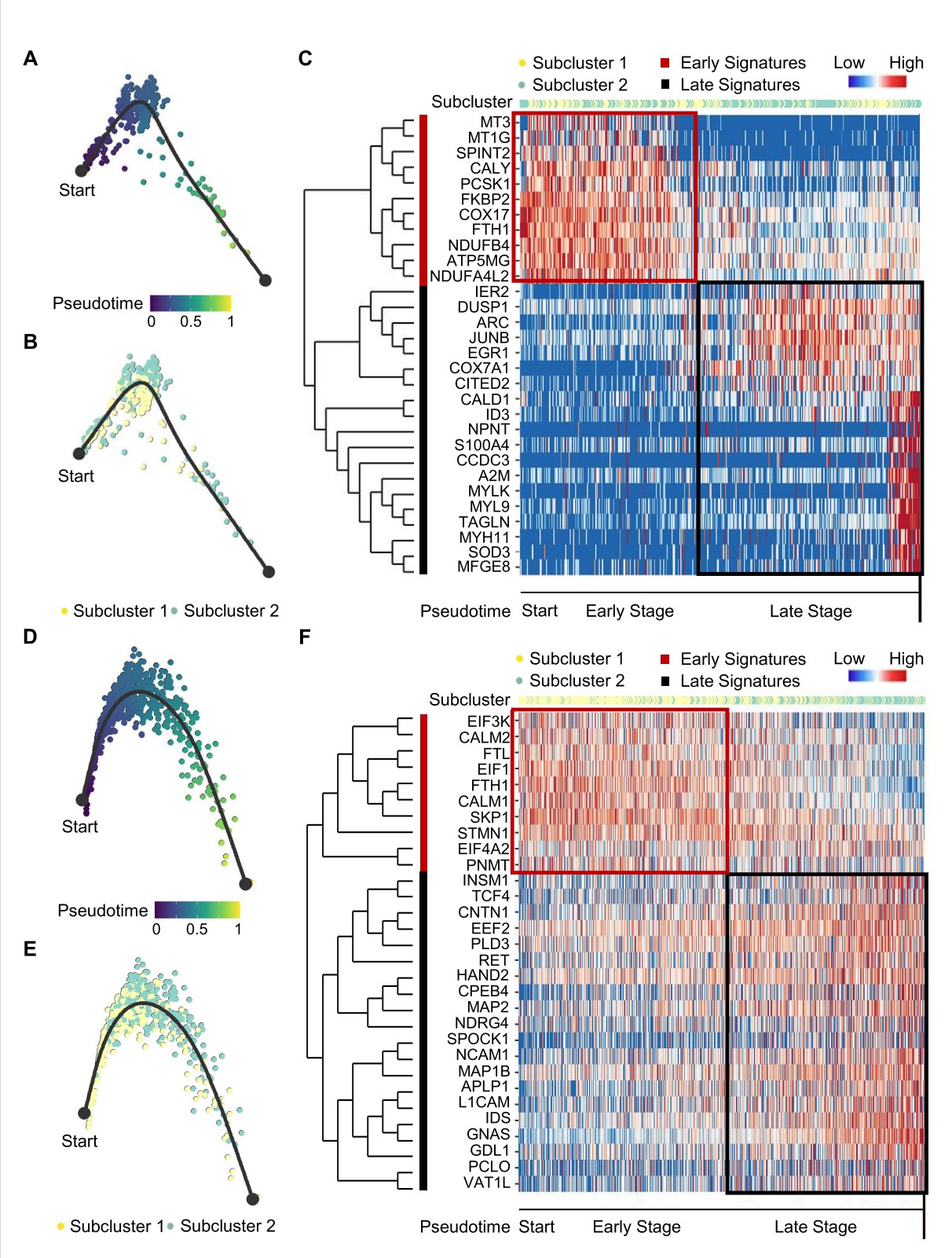

**Figure 5.** Pseudotime analysis of PCC tumor evolution by Dynverse. (**A, B**) Pseudotime trajectory of metabolism-type PCC cells colored by pseudotime (**A**) and tumor subclusters (**B**). (**C**) Pseudotime heatmap plotting the expression levels of genes across the transition from beginning (left) to end (right). The color keys from blue to red indicate the gene expression levels from low to high. (**D, E**) Pseudotime trajectory of kinase-type PCC cells colored by pseudotime (**D**) and tumor subclusters (**E**). (**F**) Pseudotime heatmap showing the expression levels of genes across the transition from beginning (left) to end (right). The color keys from blue to red indicate the gene expression levels from low to high.

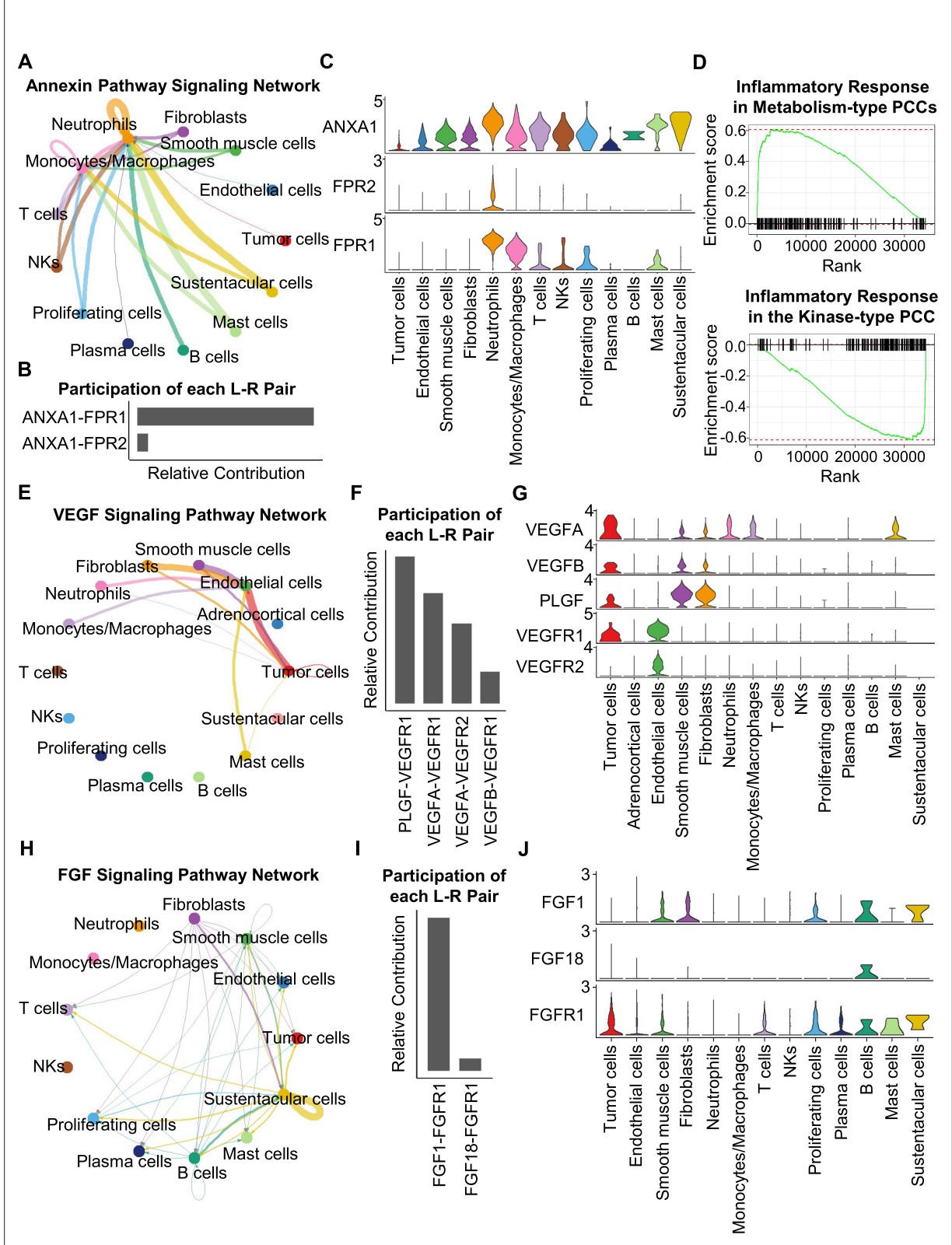

**Figure 6.** CellChat analysis reveals cell-cell communication patterns in PCC types. (**A**) Circle plot showing an inferred annexin signaling pathway network in the metabolism-type PCC microenvironment. The edges connecting the circles represent the communication probability between any two kinds of cell types. The color of the edge denotes directionality (i.e., senders *vs* receivers). (**B**) Bar graph plotting the quantification of the relative contributions of individual ligand-receptor pairs to the overall annexin signaling pathway. (**C**) Violin plot of the expression distribution of *ANXA1*, *FPR1*, and *FPR2* in

*Figure 6 continued on next page*

*Figure 6 continued*

main cell types of the metabolism-type PCC microenvironment. (**D**) GSEA enrichment plots of the inflammatory response signaling of metabolism-type patients (left) and the kinase-type patient (right). (**E**) Circle plot depicting an inferred VEGF signaling pathway. (**F**) Bar graph displaying the quantification of the relative contributions of individual ligand-receptor pairs to the overall VEGF communication network. (**G**) Violin plot plotting the expression distribution of *VEGFA*, *VEGFB*, *PLGF*, *VEGFR1*, and *VEGFR2* in main cell types of P5 PCC microenvironment. (**H**) Circle plot of an inferred FGF signaling pathway in the kinase-type PCC microenvironment. (**I**) Bar graph showing the quantification of the relative contributions of individual ligand-receptor pairs to the overall FGF signaling pathway. (**J**) Violin plot plotting the expression distribution of *FGF1*, *FGF18*, and *FGFR1* in main cell types of the kinase-type PCC microenvironment.

The online version of this article includes the following figure supplement(s) for figure 6:

**Figure supplement 1.** The frequency distribution of cell types within the microenvironment of metabolism-type and kinase-type PCC patients.

Taken together, metabolism-type PCCs exhibited a strong inflammatory reaction and the activation of annexin signaling pathway.

Considering the heterogeneity in cell type composition between sporadic and VHL patients in metabolism-type, we further analyzed inter-cellular interactions in the VHL patient. We noticed the activation of VEGF signaling pathway in P5 (*Figure 6E*), and major contribution of *PLGF-VEGFR1*, *VEGFA-VEGFR1* and *VEGFA-VEGFR2* pairs to VEGF signaling pathway (*Figure 6F*). Specifically, *VEGFA* and *VEGFR1* were highly expressed in tumor cells, *VEGFR1* and *VEGFR2* in endothelial cells, and *PLGF* in fibroblasts and smooth muscle cells, respectively (*Figure 6G*), suggesting the crosstalk between tumor cells and fibroblasts or endothelial cells in tumor microenvironment through these ligand-receptor pairs, which mediated in the tumor angiogenesis and metastasis. The observed activation of VEGF pathway is consistent with the vascular endothelial hyperplasia caused by *VHL* mutation in VHL syndrome (*Vortmeyer et al., 2013*) and the elevated proportion of endothelial cells in this patient (*Figure 2D*).

We subsequently analyzed the kinase-type PCC patient, and identified the activation of FGF signaling network (*Figure 6H*) and major involvement of *FGF1-FGFR1* in the FGF pathway (*Figure 6I*). The smooth muscle cells, fibroblasts, and B cells exhibited high expression of *FGF1*, while tumor cells, smooth muscle cells, T cells, and B cells highly expressed *FGFR1* (*Figure 6J*). The analyses suggested the cell-cell communication among tumor cells, stroma cells, and immune cells through the *FGF1-FGFR1* pair, which regulated the tumor cell proliferation (*Katoh, 2016*). These results indicate that kinase-type PCCs may benefit from FGFR inhibitors, which have been shown to exert inhibitory effects on tumor proliferation (*Liang et al., 2012*). Together, we described the inter-cellular communication networks in microenvironment of metabolism-type and kinase-type PCCs.

## Immune infiltration in the PCC microenvironment implies patient responses to immunotherapy

We further analyzed the immune microenvironment of metabolism-type and kinase-type PCCs. Visualization of inter-cellular communication showed apparent crosstalk between tumor cells and T cells in metabolism-type patients, while we were surprised to find a lack of apparent communication between tumor cells and T cells in the kinase-type patient (*Figure 7A*). These results suggested that kinase-type PCC cells could evade immune surveillance of T cells and further indicated the immune escape potential of kinase-type PCCs. Considering leukocyte antigen-class I (HLA-I) surface level is known to represent the antigen-presenting abilities of tumor cells (*Brea et al., 2016*; *Jhunjhunwala et al., 2021*; *Oh et al., 2019*), we assessed *HLA-I* expression in PCC cells: the expression levels of *HLA-A*, *HLA-B*, and *HLA-C* were significantly lower in kinase-type PCC cells than in metabolism-type PCC cells (*Figure 7B*). Consistently, immunohistochemistry staining of serial sections of tumor tissues showed lower expression of HLA-A in P4 as compared to that in other patients (*Figure 7C*), providing a plausible explanation for the lack of interactions detected between tumor cells and T cells in the kinase-type PCC. In short, we discovered that the kinase-type PCC displayed an impaired *HLA-I* expression profile consistent with immune escape, while metabolism-type PCCs showed antigen-presenting abilities.

We additionally assessed the immune microenvironment of PCCs by subclustering immunotherapy-related immune cells and identified eight clusters including central memory CD4[+] T cells (CD4[+] TCM, marked by *CD3D*, *CD3E*, *CD3G*, *CD4*, *CCR7*, and *IL7R*), effector memory CD4[+] T cells (CD4[+] TEM, marked by *CD3D*, *CD3E*, *CD3G*, *CD4*, and *IL7R*), CD8[+] T cells (marked by *CD3D*, *CD3E*, *CD3G*, *CD8A*

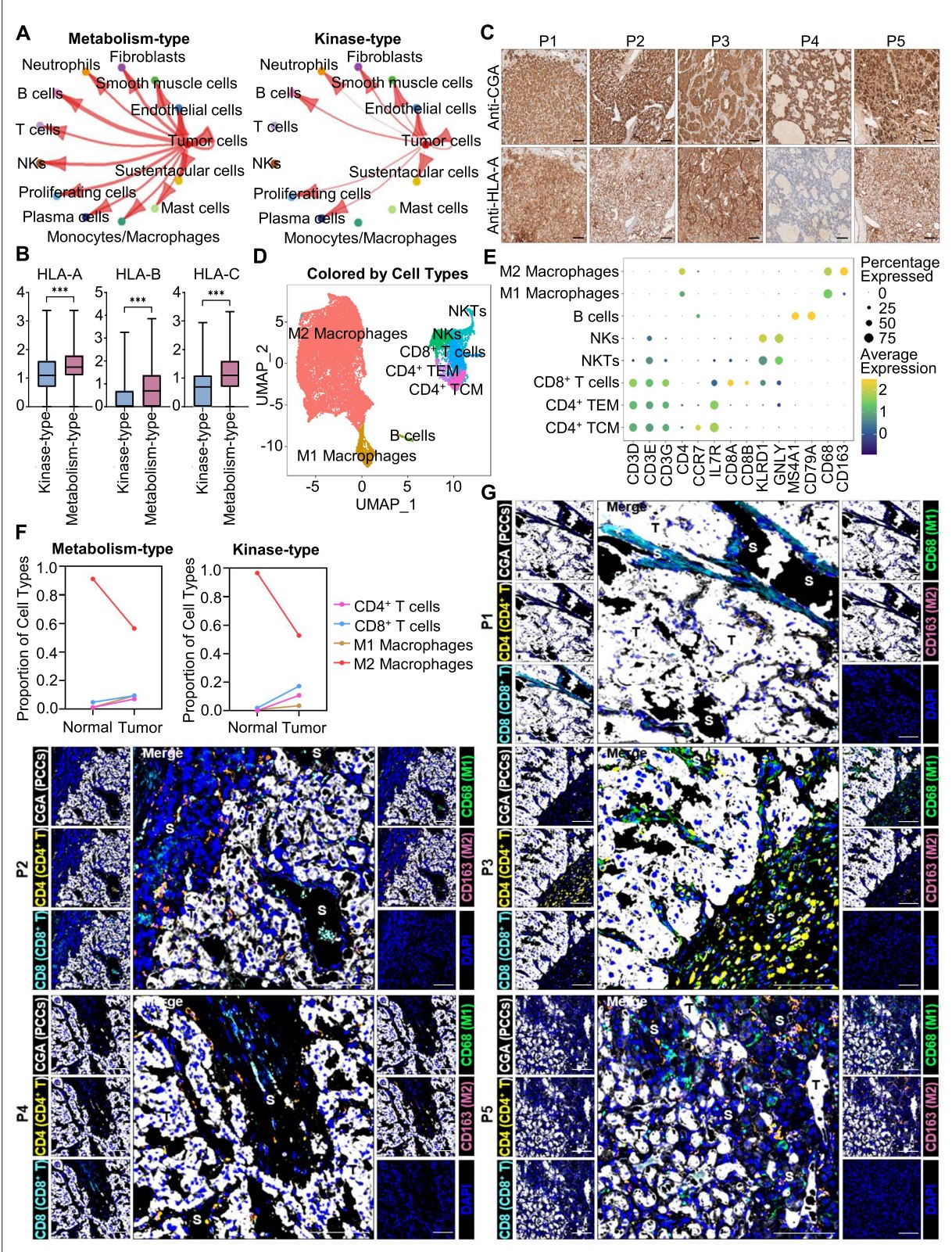

**Figure 7.** Prediction of the immune escape potential and immunotherapy response of PCC patients. (**A**) Circle plots depicting inferred inter-cellular interactions in metabolism-type (left) and kinase-type (right) PCC microenvironment. (**B**) Box plots showing the expression levels of *HLA-A*, *HLA-B*, and *HLA-C* in metabolism-type and kinase-type PCC patients. (**C**) Immunohistochemistry staining of CGA and HLA-A in formalin-fixed paraffin-embedded PCC tissue sections matched to scRNA-seq specimens. Scale bar, 100 μm. (**D**) UMAP plot showing 8 immune cell types detected from all

*Figure 7 continued on next page*

*Figure 7 continued*

PCC specimens. (**E**) Dot plot of representative marker genes for each immune cell type. The color scale represents the average marker gene expression level; dot size represents the percentage of cells expressing a given marker gene. (**F**) Comparison of the proportion of immune cell types in tumor *vs* adjacent normal adrenal medullary tissues. (**G**) Multispectral immunofluorescent staining for the juxtaposition of PCC cells (marked by CGA), CD4+ T cells (marked by CD4), CD8+ T cells (marked by CD8), M1 macrophages (marked by CD68), and M2 macrophages (marked by CD163) in formalin-fixed paraffin-embedded PCC tissue sections matched to scRNA-seq specimens. The white, yellow, cyan, green, pink, and blue spots indicated cells with high expression of CGA, CD4, CD8, CD68, CD163, and DAPI proteins in PCC tissue sections, respectively. S, stroma; T, tumor. Scale bar, 100 μm.

and *CD8B*), natural killer T cells (NKTs, marked by *CD3E*, *KLRD1* and *GNLY*), natural killer cells (NKs, marked by *KLRD1* and *GNLY*), B cells (marked by *MS4A1* and *CD79A*), M1 macrophages (marked by *CD68*), and M2 macrophages (marked by *CD68* and *CD163*) (*Figure 7D and E*). Further analysis of immune cell type composition in metabolism-type and kinase-type PCCs showed an increase in the proportion of CD4+ T cells, CD8+ T cells, and M1 macrophages in tumor tissues as compared with that in adjacent normal adrenal medullary tissues, while the proportion of M2 macrophages in adjacent normal adrenal medullary tissues decreased as compared with that in tumor tissues (*Figure 7F*).

The proportion and location of infiltrating immune cells in tumor tissues have been shown to be informative regarding responsivity to immunotherapy (*Sade-Feldman et al., 2018*; *Stanton and Disis, 2016*). In particular, the CD8+ T cells play an essential role in anti-tumor immunity; these cells can recognize tumor antigens displayed on the surface of tumor cells by HLA-I molecules (*Jhunjhunwala et al., 2021*). As the scRNA-seq data do not support discernment of whether immune cells have infiltrated tumors or are merely present in the sampled materials, we performed multispectral immunofluorescence (mIF) staining on CD4+ T cells, CD8+ T cells, M1 macrophages, and M2 macrophages in formalin-fixed paraffin-embedded (FFPE) tumor tissues of P1-P5. We observed that CD8+ T cells, as well as CD4+ T cells, M1 macrophages, and M2 macrophages were rare (and were only present in tumor stroma of P1-P5), indicating apparent immune escape in both metabolism-type and kinase-type PCCs (*Figure 7G*). Combined with previously reported negative regulatory effects of kinases (such as RET, ALK, and MEK) on HLA-I expression on tumor cells (*Brea et al., 2016*; *Oh et al., 2019*), we speculate that the possible reason for inability in recruiting CD8+ T cells of kinase-type PCCs is the down-regulation of *HLA-I* in tumor cells regulated by *RET*, while the mechanism of immune escape in metabolism-type PCCs (with antigen presentation ability) needs to be further explored. Our results also indicate that the application of immunotherapy to metabolism-type PCCs is likely unsuitable, while kinase-type PCCs may have the potential of combined therapy with kinase inhibitors and immunotherapy.

## Discussion

The genomic research of PCCs has made continuous progress in recent years (*Bausch et al., 2017*; *Mete et al., 2022*; *Neumann et al., 2019*; *Nölting et al., 2022*), but there have been obvious breakthroughs for more efficacious therapies. Isotope therapies can only relieve the clinical symptoms in 50% of patients (without reducing tumor size; *Fitzgerald et al., 2006*; *Roman-Gonzalez and Jimenez, 2017*), while most chemotherapies and targeted therapies exhibit low remission rates and severe side effects in clinical studies (*Druce et al., 2009*; *Hamidi, 2019*; *O'Kane et al., 2019*; *Oh et al., 2012*), limiting the development of PCC treatments. Although immunotherapy has achieved success in the therapy of solid tumors, it is difficult to evaluate the curative effect in PCCs. To address this problem, two key features associated with the tumor immunotherapy need to be elucidated—tumor heterogeneity and the tumor microenvironment (*Hegde and Chen, 2020*; *Li et al., 2018*; *Liu et al., 2023*; *Liu et al., 2022*; *Zhang and Zhang, 2020*). Regarding PCC heterogeneity, genomic studies have confirmed the inter-individual heterogeneity based on driver mutations (*Dahia, 2014*; *Jhawar et al., 2022*; *Sarkadi et al., 2022*; *Toledo et al., 2017*). However, there is lack of relevant research on intratumoral heterogeneity, which has an impact on disease progression and sensitivity to immunotherapy (*Vitale et al., 2021*). For the PCC microenvironment, the cell composition and inter-cellular crosstalk remain largely unexplored. By investigating the heterogeneity and microenvironment characteristics of PCCs, we proposed a feasible method to judge the heterogeneity of PCCs, and also provided a clue for the application of immunotherapy in PCC patients.

The clinical and histopathological diagnosis of metastasis risk and heterogeneity of PCCs is particularly difficult, and is still limited by the lack of reliable prognostic markers (*de Wailly et al., 2012*). Although the PASS scoring system was proposed as a tool for discriminating potentially metastasis PCCs from benign ones, its utility is somewhat restricted due to observer variation and nonrepeatability (*Agarwal et al., 2010*; *Kimura et al., 2014*; *Thompson, 2002*; *Wu et al., 2009*). By integrating the PASS scoring system with scRNA-seq analysis, we observed that both methods offered complementary insights into the intra-tumoral and inter-individual heterogeneity of PCCs. Considering the intricate cellular components within the PCC microenvironment and recognizing the advantage of scRNA-seq in dissecting the tumor microenvironment (*Ren et al., 2021*), the potential integration of molecular diagnostic methods, such as single-cell sequencing, with pathological tools appears promising for implementation in the clinical practice of PCC diagnosis and care (*Crona et al., 2017*; *Papathomas et al., 2021*).

According to the results of genomics research in recent years, 40~50% of PCCs have been classified into a certain molecular pathway (pseudohypoxia, kinase, or Wnt), but the remaining 50–60% are still unclassified (*Crona et al., 2017*; *Lenders et al., 2014*). PCCs that classified into these molecular pathways may have the opportunity to carry out drug clinical trials against corresponding targets, such as HIF2α, VEGF, and RET, etc. (*Jhawar et al., 2022*; *Nölting et al., 2019*; *Toledo and Jimenez, 2018*), while unclassified PCCs still have no clues or basis for drug treatment. According to the WES results of the five patients in our study, except for one patient with *VHL* germline mutation, the others could not be classified based on detected germline or somatic mutations. Therefore, we developed a new classification of PCCs based on scRNA-seq, which are able to accurately capture the transcriptional features of PCC cells. We found that PCC cells from four patients highly expressed metabolism-related genes, while that from another patient exhibited high expression of kinase-related genes. Although previous studies applying bulk RNA-seq analysis of PCCs have also reported highly expressed genes (*Batchu et al., 2022*; *Flynn et al., 2015*), the interpretation of bulk RNA-seq data can be complicated by interference from the large number of non-tumor cells (*Huang et al., 2023*; *Li et al., 2022*), so it is difficult to determine whether the highly expressed genes are reliable molecular features of PCCs. Our study revealed that the kinase-type PCC patient (P4) exhibited higher blood pressures and plasma levels of catecholamine metabolites (3-methoxytyramine and normetanephrine). Further research is warranted to explore the correlation of our molecular classification with plasma levels of catecholamine metabolites.

The tumor immune microenvironment, defined as the interplay between tumor cell antigen presentation, immune cell infiltration, and their interactions, can be multifaceted (*de Visser and Joyce, 2023*; *Hanahan, 2022*). Given the spatial limitations of scRNA-seq and the protein detection constraints of multispectral immunofluorescent staining, we combined these two methods to investigate immune escape mechanisms in both types of PCCs. Although our immunofluorescence staining data showed a lack of CD8$^+$ T cell infiltration in both metabolism-type and kinase-type PCCs, only kinase-type PCCs exhibited downregulation of *HLA-I* molecules. The expression of HLA-I is a marker for tumor antigen presentation and CD8$^+$ T cell infiltration (*Jhunjhunwala et al., 2021*; *Perea et al., 2018*; *Sadagopan et al., 2022*). Previous studies have demonstrated that kinases such as RET, MAP2K1, ALK, and FGF can promote tumor growth via helping tumor cells to evade the immune system by downregulating HLA-I expression (*Brea et al., 2016*; *Oh et al., 2019*). The kinase-type PCCs that we defined exhibited activation of RET and FGF signals, which led us to consider the possible inhibitory effect of kinases on HLA-I expression in PCCs. Recently, RET inhibitors have been approved for the treatment of non-small cell lung cancer with *RET* mutation (*Drilon et al., 2020*; *Griesinger et al., 2022*). For kinase-type PCCs, whether RET inhibitors can enhance the antigen presentation ability of tumor cells, and further provide chance for combination of kinase-targeted therapy and immunotherapy, needs to be verified by follow-up research. The metabolism-type PCCs, however, with antigen presentation ability and also the lack of CD8$^+$ T cell infiltration, the application of immunotherapy is likely unsuitable, and the mechanism(s) of immune escape in these tumors needs to be further explored.

There are potential limitations of our study. Despite the patient cohort in our study including a hereditary PCC patient, only one patient with a *VHL* germline mutation was included; other types of hereditary PCCs with germline mutations (for example in *SDHx* and *FH*) were not included. The number of sporadic cases was also limited, mainly caused by the low incidence of PCCs. In addition,

the specimens were all from primary tumors, while no metastatic tumors were included. Together, these considerations underscore the necessary for additional studies of PCCs in the future.

In summary, our study presents the molecular classification and tumor microenvironment characterization of PCCs through scRNA-seq. The intra-tumoral heterogeneity is lower than the inter-individual heterogeneity of PCCs. Among unclassified PCCs, we defined metabolism-type and kinase-type at the single-cell transcriptome level. We observed a lack of CD8[+] T cell infiltration in both metabolism-type and kinase-type PCCs. The kinase-type PCC showed downregulation of *HLA-I* that possibly regulated by *RET*, suggesting the potential of combined therapy with kinase inhibitors and immunotherapy in kinase-type PCCs. For the metabolism-type PCCs, which have antigen presentation ability and also exhibit immune escape, the application of immunotherapy is likely unsuitable. The proposal of this molecular classification and our characterization of these PCC types can contribute to strategy development and clinical trials of PCC treatments in the future.

## Methods

### Ethical regulations
The research presented here complies with all relevant local, national, and international regulations. For all PCC patient specimens, informed written consent was obtained prior to donation. The Peking University First Hospital Review Board (Protocol 300–001) approved the study.

### Patient cohort
Five PCC patients were included, and all patients had signed the consent forms at the Department of Urology in Peking University First Hospital. Fresh tumor specimens were collected during surgical resection. The sporadic patients (P1-P4) were performed surgical resections of the tumors at right adrenals, and the VHL patient (P5) underwent left adrenalectomy. We collected 1 resected tumor specimen from P1 (P1_T1), 3 specimens from distinct intra-tumoral sites from P2 (P2_T1, P2_T2, and P2_T3) or P5 (P5_T1, P5_T2, and P5_T3), and 2 specimens from distinct intra-tumoral sites from P3 (P3_T1 and P3_T2), or P4 (P4_T1 and P4_T2). To enable comparisons, one normal adrenal medullary tissue adjacent to the tumor was collected from each patient (P1_A, P2_A, P3_A, P4_A, and P5_A). A total of sixteen specimens were carefully dissected under the microscope and confirmed by a qualified pathologist.

### Tissue processing
The fresh tumor specimens were stored in the tissue preservation solution (JSENB) and washed with Hanks Balanced Salt Solution (HBSS, HyClone) for three times and minced into 1–2 mm pieces. Then the tissue pieces were digested with 2 ml tissue dissociation solution at 37°C for 15 min in 15 ml centrifuge tube with sustained agitation. After digestion, using 40 μm sterile strainers to filter the samples and centrifuging the samples at 1,000 rpm for 5 min. Then the supernatant was discarded, and the sediment was resuspended in 1 ml phosphate buffered saline (PBS, Solarbio). To remove the red blood cells, 2 ml red blood cell lysis buffer (BD) was added at 25 °C for 10 min. The solution was then centrifuged at 500 × *g* for 5 min and suspended in PBS. The cells were stained with trypan blue (Sigma) and microscopically evaluated.

### Single-cell library construction and sequencing
Utilizing the 10x Genomics Chromium Single Cell 3′ v3 Library Kit and Chromium instrument, approximately 17,500 cells were partitioned into nanoliter droplets to achieve single-cell resolution for a maximum of 10,000 individual cells per sample. The resulting cDNA was tagged with a common 16 nt cell barcode and 10 nt Unique Molecular Identifier during the RT reaction. Full-length cDNA from poly-A mRNA transcripts was enzymatically fragmented and size selected to optimize the cDNA amplicon size (approximately 400 bp) for library construction (10x Genomics). The concentration of the 10x single-cell library was accurately determined through qPCR (Kapa Biosystems) to produce cluster counts appropriate for the HiSeq4000 or NovaSeq6000 platform (Illumina). In all, 26×98 bp (3′ v3 libraries) sequence data were generated targeting between 25 and 50 K read pairs/cell, which provided digital gene expression profiles for each individual cell.

## General scRNA-seq data analysis

For single-cell RNA-seq analysis, we used CellRanger (10x Genomics, v.6.1.2) to pre-process the single-cell RNA-seq data after obtaining the paired-end raw reads. Cell barcodes and unique molecular identifiers (UMIs) of the library were extracted from read 1. Then, the reads were split according to their cell (barcode) IDs, and the UMI sequences from read 2 were simultaneously recorded for each cell. Quality control on these raw readings was subsequently performed to eliminate adapter contamination, duplicates, and low-quality bases. After filtering barcodes and low-quality readings that were not related to cells, we mapped the cleaned readings to the human genome (GRCh38) and retained the uniquely mapped readings for UMIs counts. Next, we estimated the accurate molecular counts and generated a UMI count matrix for each cell by counting UMIs for each sample. Finally, we generated a gene-barcode matrix that showed the barcoded cells and gene expression counts.

The R package Seurat (v.4.0.2) was used for all subsequent analysis (*Butler et al., 2018*; *Satija et al., 2015*; *Stuart et al., 2019*). For quality control of single-cell RNA-seq, a series of quality filters was applied to the data to remove those barcodes which fell into any one of these categories: too few genes expressed (possible debris), too many UMIs associated (possible more than one cell), and too high mitochondrial gene expression (possible dead cell). The cutoffs for these filters were as follows: the minimum number (no less than 200) and maximum number (no more than 5000) were used in controlling the number of genes; the maximum number (no more than 30%) was used for the quality control of the percentage of mitochondrial genes. Low-quality cells and outliers were discarded, and the single cells that passed the QC criteria were used for downstream analyses.

## Cell clustering and cell type annotation

The Seurat software package (v.4.0.2) was used to perform cell clustering analysis to identify major cell types (*Butler et al., 2018*; *Satija et al., 2015*; *Stuart et al., 2019*). All Seurat objects constructed from the filtered UMI-based gene expression matrixes of given samples were merged. We first applied the SCTransform (v.0.3.2) function to implement normalization, variance stabilization, and feature selection through a regularized negative binomial model. Then, the principal component analysis (PCA) was applied for linear dimensionality reduction with the top 3000 variable genes. According to standard steps implemented in Seurat, highly variable numbers of principal components (PCs) 1–50 were selected and used for clustering using the Uniform Manifold Approximation and Projection (UMAP) method. We identified cell types of these clusters based on the expression of canonic cell type markers or inferred by CellMarker database (*Zhang et al., 2019*). The cluster markers were also certified using the FindAllMarkers function of the Seurat suite, and cell types were manually annotated based on certified markers finally.

## Correlation analysis of scRNA-seq

After integration, for each cluster, each sample, and each patient, we compared the gene expression to others to identify the significant highly expressed genes (adjusted p-value < 0.05 and log fold change > 0). Then the average gene expressions in each cluster, each sample, and each patient were calculated. The pairwise correlations were then estimated.

## DEGs identification and enrichment analysis

The cluster-specific genes were identified by running Seurat containing the function of FindAllMarkers on a log-transformed expression matrix (min. pct = 0. 25, only. Pos = TRUE, and logfc.threshold = 0.25). We also identified the differentially over-expressed genes between two clusters with the Wilcoxon Rank-Sum Test with the FindMarkers function in Seurat (adjusted p-value < 0.05, only.pos = T, and logfc.threshold = 0.1), and the cluster-specific overrepresented GO biological process was calculated with the compareCluster function in the clusterProfiler package (v.4.2.2) of R (*Yu et al., 2012*). We also used the GSEA with the curated gene sets to identify the pathways that were induced or repressed in between the cell clusters. In brief, the mean gene expression level was calculated and the log twofold change (FC) between the specific cell cluster and the other cells was applied as the test statistic. The 50-hallmark gene sets in the MSigDB databases (https://www.gsea-msigdb.org/gsea/msigdb) were used for the GSEA analysis (*Liberzon et al., 2011*).

### Single-cell CNV detection and clustering

All cells were classified as either normal or tumor based on the genome-wide copy number profiles computed from the gene expression UMI matrix using the Bayesian segmentation approach, CopyKat (v.1.0.8; *Gao et al., 2021*). Aneuploid single cells with genome-wide copy number aberrations were anticipated to be tumor cells, while diploid cells were presumed to be normal cells. The CopyKat-based predictions were subsequently validated by single-cell gene expression profiles.

### Trajectory analysis of scRNA-seq data

In order to perform a detailed comparison among different trajectory modeling tools, the Dynverse (v.0.1.2) tool was used (*Saelens et al., 2019*). Based on the scoring system provided by Dynverse and following a careful inspection of the generated trajectories, we applied 'SCORPIUS' and 'EMBEDDR' to infer the development trajectories. Finally, we selected the genes that were differentially expressed on different stages through the trajectory and plotted the pseudotime heatmap.

### Cell-cell communication analysis

Cell-cell interactions based on the expression of known ligand-receptor pairs in different cell types were inferred using CellChatDB (v.1.1.3) (*Jin et al., 2021*). In brief, we followed the official workflow and loaded the normalized counts into CellChat and applied the preprocessing functions 'identify-OverExpressedGenes', 'identifyOverExpressedInteractions', and 'projectData' with standard parameters set. As database we selected the 'Secreted Signaling' pathways and used the pre-compiled human 'Protein-Protein-Interactions' as a priori network information. For the main analyses, the core functions 'computeCommunProb', 'computeCommunProbPathway', and 'aggregateNet' were applied using standard parameters and fixed randomization seeds. Finally, to determine the senders and receivers in the network, the function 'netAnalysis_signalingRole' was applied on the 'netP' data slot.

### Whole exome sequencing and analysis

Genomic DNA extracted from tumor tissues were sent for whole exome sequencing. The exomes were captured using the Agilent SureSelect Human All Exon V6 Kit and the enriched exome libraries were constructed and sequenced on the Illumina NovaSeq 6000 platform to generate WES data (150 bp paired-end reads, > 100 × ) according to standard manufacturer protocols. The cleaned reads were aligned to the human reference genome sequence UCSC Build 19 (hg19) using Burrows-Wheeler Aligner (BWA) (v.0.7.12; *Li and Durbin, 2009*). All aligned BAM were then performed through the same bioinformatics pipeline according to GATK Best Practices (v.3.8; *McKenna et al., 2010*). We obtained germline variants based on variant calling from GATK-HaplotypeCaller. We then used GATK-MuT ect2 to call somatic variants in tumors and obtained a high-confidence mutation set after rigorous filtering by GATK-FilterMutectCalls. All variants were annotated using ANNOVAR (v.2018Apr16; *Wang et al., 2010*).

### Immunocytochemistry and multispectral immunofluorescent staining

Immunocytochemistry and multispectral immunofluorescent staining experiments were conducted according to standard protocols using antibodies against formalin-fixed paraffin-embedded (FFPE) tissue specimens. The antibodies used are listed as follows: CGA (ABclonal, A9576), NDUFA4L2 (Proteintech, 66050–1-lg), COX4I2 (Santa, sc-100522), PNMT (Abcam, ab154282), RET (Abcam, ab134100), CD4 (Abcam, ab133616), CD8 (Invitrogen, MA1-80231), CD68 (Invitrogen, MA5-12407), CD163 (Abcam, ab182422), and HLA-A (ABclonal, A11406).

## Acknowledgements

This work was supported by grant (No. 2021YFA1300603 to LS) from the Ministry of Science and Technology of China, grant (31991164, 82188102, 32350020, and 32370620 to LS, 82141103 and 82172617 to KG) from the National Natural Science Foundation of China, grant (Z200020 to LS) from the Natural Science Foundation of Beijing, grant (2022-2-4074 to KG) from Capital's Funds for Health Improvement and Research, and grant (2022CX08 to ZZ and 2022CR75 to KG) from National High-Level Hospital Clinical Research Funding. We thank the National Center for Protein Sciences at

Peking University (Beijing, China) and Dr. Jiaqian Wang (YuCe Biotech Co., Ltd) for providing technical support.

## Additional information

### Funding

| Funder | Grant reference number | Author |
|---|---|---|
| Ministry of Science and Technology of the People's Republic of China | No. 2021YFA1300603 | Luyang Sun |
| National Natural Science Foundation of China | 31991164 | Luyang Sun |
| National Natural Science Foundation of China | 82188102 | Luyang Sun |
| National Natural Science Foundation of China | 32350020 | Luyang Sun |
| National Natural Science Foundation of China | 32370620 | Luyang Sun |
| National Natural Science Foundation of China | 82141103 | Kan Gong |
| National Natural Science Foundation of China | 82172617 | Kan Gong |
| Natural Science Foundation of Beijing Municipality | Z200020 | Luyang Sun |
| Capital's Funds for Health Improvement and Research | 2022-2-4074 | Kan Gong |
| National High-Level Hospital Clinical Research Funding | 2022CX08 | Zheng Zhang |
| National High-Level Hospital Clinical Research Funding | 2022CR75 | Kan Gong |

The funders had no role in study design, data collection and interpretation, or the decision to submit the work for publication.

### Author contributions

Sen Qin, Software, Formal analysis, Visualization, Methodology, Writing - original draft; Yawei Xu, Data curation, Methodology; Shimiao Yu, Wencong Han, Software, Methodology; Shiheng Fan, Wenxiang Ai, Visualization, Methodology; Kenan Zhang, Yizhou Wang, Software, Visualization; Xuehong Zhou, Supervision, Methodology; Qi Shen, Software, Visualization, Methodology; Kan Gong, Luyang Sun, Zheng Zhang, Supervision, Funding acquisition, Writing – review and editing

### Author ORCIDs

Sen Qin ![ORCID] http://orcid.org/0000-0002-1356-453X
Yawei Xu ![ORCID] http://orcid.org/0000-0002-5129-186X
Luyang Sun ![ORCID] http://orcid.org/0000-0003-3917-473X

### Ethics

Human subjects: The research presented here complies with all relevant local, national, and international regulations. For all PCC patient specimens, informed written consent was obtained prior to donation. The Peking University First Hospital Review Board (Protocol 300-001) approved the study.

Reviewer #2 (Public Review): https://doi.org/10.7554/eLife.87586.3.sa1

Reviewer #3 (Public Review): https://doi.org/10.7554/eLife.87586.3.sa2
Author Response https://doi.org/10.7554/eLife.87586.3.sa3

## Additional files

### Supplementary files
• Supplementary file 1. WES Detection, PASS Scores, and Clinical Information of 5 PCC Patients. (a) Somatic and Germline Mutations in 5 PCC Patients Detected by WES. (b) PASS Scores of Collected Tumor Tissues. (c) Clinical Information of 5 PCC Patients.

• MDAR checklist

### Data availability
All data generated or analyzed during this study are included in the manuscript and supplementary files; human transcriptome reference used for our analysis is available at 10 x Genomics website (https://cf.10xgenomics.com/supp/cell-exp/refdata-cellranger-GRCh38-3.0.0.tar.gz) or in zenodo (https://zenodo.org/record/4114854/files/refdata-cellranger-GRCh38-3.0.0.tar.gz?download=1). The single-cell RNA sequencing data was processed using cellranger (v.6.1.2) and analyzed with the R package Seurat (v.4.0.2); the code used for scRNA-seq analysis is available at 10x Genomics website (https://www.10xgenomics.com/support/software/cell-ranger/latest/analysis/running-pipelines/cr-gex-count) and at GitHub website (https://github.com/satijalab/seurat/releases/tag/v4.0.2; *Satijalab, 2021*).

The following dataset was generated:

| Author(s) | Year | Dataset title | Dataset URL | Database and Identifier |
|---|---|---|---|---|
| Sen Q | 2024 | Microenvironment Characteristics and Molecular Classification in Pheochromocytoma Patients | https://doi.org/10.5061/dryad.rjdfn2zkg | Dryad Digital Repository, 10.5061/dryad.rjdfn2zkg |

The following previously published dataset was used:

| Author(s) | Year | Dataset title | Dataset URL | Database and Identifier |
|---|---|---|---|---|
| Rabilloud T, Delphine P, Saran P, Mathis N, Loosveld M, Payet-Bornet D | 2020 | Single-cell profiling identifies pre-existing CD19-negative subclones in a B-ALL patient with CD19-negative relapse after CAR-T therapy | https://doi.org/10.5281/zenodo.4114854 | Zenodo, 10.5281/zenodo.4114854 |

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
