## [Editor Report · eLife assessment]

This **valuable** study advances our understanding of the potential therapeutic strategies for the treatment of pheochromocytomas using single-cell transcriptomics. The authors propose a new molecular classification criterion based on the characterization of tumor microenvironmental features, based on **solid** evidence. The work, which could be improved further through delineating the choice of the PASS scoring system, will be of broad interest to clinicians, medical researchers, and scientists working in the field of pheochromocytoma.

---

## [Referee Report · Reviewer #2 (Public Review)]

Pheochromocytoma (PCC), a rare neuroendocrine tumor, is currently considered malignant, but non-surgical treatment options are very limited and there is an urgent need for more basic research to support the development of new therapeutic approaches. In the present work, the authors described the intra- and inter-tumor heterogeneity by performing scRNA-seq on tumor samples from five patients with PCC, and evaluated the corresponding PASS scores.

Strengths: The tumor microenvironment of PCC was characterized and potential molecular classification criteria based on single-cell transcriptomics were proposed, offering new theoretical possibilities for the treatment of PCC. The article is logically written and the results are clearly presented.

Weaknesses: I still have concerns about some of the article's content. My main concerns are: In this study, the authors seem to have demonstrated the inaccuracy of a subjective score (PASS) by another objective means (scRNA-seq). In fact, the multiparametric scoring systems such as PASS are no longer endorsed in the 2022 WHO guidelines. The PASS scoring system does not have a high positive predictive value for risk stratification of PCC metastasis, but "rule-out" of metastasis risk with a PASS score of <4 seems to be fairly reliable. Could the authors please explain why the PASS scores were chosen rather than the GAPP, m-GAPP, or COPPS scoring systems? If possible, please try to emphasize the importance and necessity of using the PASS scoring system, either by replacing it with a more acceptable scoring system or by deleting the relevant part, which does not seem to be very relevant to the subject of the article.

Moreover, I noted the following statement in the text "There are no studies reporting the composition of immune cells in PCCs. The few published studies investigating the immune microenvironment of PCCs have been limited to the expression of PDL1 at the histological level and to assessment of the tumor mutation burden (TMB) at the genomic level, and these results only seem to suggest that PCCs are immune-cold (Bratslavsky et al, 2019; Guo et al, 2019; Pinato et al, 2017)." This statement is very wrong. The reason for this error may be that the authors did not adequately search and read the relevant literature. I noticed that almost all references in this paper are dated 2021 and earlier, which is surprising. Please update the references cited in this paper in a comprehensive and detailed manner; referring to literature published too early may lead to inadequate discussion or even one-sided or incorrect conclusions and conjectures.

For example, the text statement "Combined with previously reported negative regulatory effects of kinases (such as RET, ALK, and MEK) on HLA-I expression on tumor cells (Brea et al., 2016; Oh et al., 2019), we speculate that the possible reason for inability in recruiting CD8+ T cells of kinase-type PCCs is the downregulation of HLA-I in tumor cells regulated by RET, while the mechanism of immune escape in metabolism-type PCCs (with antigen presentation ability) needs to be further explored. Our results also indicate that the application of immunotherapy to metabolism-type PCCs is likely unsuitable, while kinase-type PCCs may have the potential of combined therapy with kinase inhibitors and immunotherapy." is rather one-sided; in fact, the presence of immune escape in PCC, as the malignancy with the lowest tumor mutation compliance, has been well characterized, and the low number of infiltrating T cells in tumor tissue may be influenced by a variety of factors, such as the release of catecholamines, the expression of inhibitory receptors on the surface of T cells, and so on, although genetic mutation still plays the most crucial role. The Discussion section also has a lot of information that needs to be updated or corrected and expanded, so please rewrite the above section with sufficiently updated references.

Below I have listed some references for the authors to read:

Tufton N, Hearnden RJ, Berney DM, et al. The immune cell infiltrate in the tumour microenvironment of phaeochromocytomas and paragangliomas. Endocr Relat Cancer. 2022;29(11):589-598. Published 2022 Sep 19. doi:10.1530/ERC-22-0020

Jin B, Han W, Guo J, et al. Initial characterization of immune microenvironment in pheochromocytoma and paraganglioma. Front Genet. 2022;13:1022131. Published 2022 Dec 7. doi:10.3389/fgene.2022.1022131

Celada L, Cubiella T, San-Juan-Guardado J, et al. Pseudohypoxia in paraganglioma and pheochromocytoma is associated with an immunosuppressive phenotype. J Pathol. 2023;259(1):103-114. doi:10.1002/path.6026

Calsina B, Piñeiro-Yáñez E, Martínez-Montes ÁM, et al. Genomic and immune landscape Of metastatic pheochromocytoma and paraganglioma. Nat Commun. 2023;14(1):1122. Published 2023 Feb 28. doi:10.1038/s41467-023-36769-6

---

## [Referee Report · Reviewer #3 (Public Review)]

The main findings of this study are as follows: (1) The authors defined "metabolism-type" and "kinase-type" in unclassified sporadic PCC patients through the single-cell transcriptomics-based differentially expressed genes and functional enrichment analyses. (2) They identified the limitation of Pheochromocytoma of the Adrenal gland Scaled Score (PASS) system and suggested the combination of molecular diagnostic methods like scRNA-seq with pathological tools like PASS in aiding the clinical evaluation of PCCs. (3) Analysis of the PCC microenvironment revealed a lack of immune cell infiltration in both metabolism-type and kinase-type PCCs, while only the kinase-type PCC patient exhibited the low expression of HLA-Ⅰ that potentially regulated by RET, providing clues for the combined therapy with kinase inhibitors and immunotherapy in kinase-type PCC patients.

The main strength of this manuscript is that it involves scRNA-seq analysis of an extremely rare tumor type-PCCs, which presents a single-cell transcriptomics-based molecular classification and microenvironment characterization of PCCs and further provides clues for potential therapeutic strategies to treat PCCs. The authors also validated the scRNA-seq analysis results (such as the expression levels of marker genes and the distribution of immune cells in the PCC microenvironment) with immunocytochemistry and multispectral immunofluorescent staining. In summary, the findings in this manuscript are quite interesting and significant, which will potentially be valuable for the molecular classification of PCCs.

---

## [Author Response]

The following is the authors’ response to the original reviews.

**eLife assessment**
This study presents a valuable finding for the treatment of PCCs by sequencing 16 tumor specimens from five patients with pheochromocytomas by single-cell transcriptomics and proposing a new molecular classification criterion based on the sequencing results and characterization of tumor microenvironmental features. The evidence supporting the claims of the authors is solid, although the inclusion of more patient samples would strengthen the study's conclusions. The work will be of interest to clinicians or medical biologists working on rare pheochromocytomas (PCCs).

Firstly, we sincerely appreciate the positive feedback from the editor and extend our gratitude to the three reviewers for their meticulous review and valuable comments. Our detailed responses to each recommendation are outlined below.

**Response to reviewers’ recommendations**

**Reviewer #1 (Recommendations for The Authors):**
1. Transcriptomal clonal dynamics of different PCCs is well written. However for conclusion sample size needs to be more.

Acknowledging the rarity of PCCs with an incidence of approximately 0.2 to 0.6 cases per 100,000 person-years (Farrugia & Charalampopoulos, 2019; Neumann et al, 2019), our study recognizes the limitation in sample size, as discussed in the limitations section (Page 22). In response to this concern, we are committed to undertaking further research with an expanded sample size to bolster the robustness of our conclusions, seeking a more comprehensive understanding of tumor microenvironment characterization and molecular classification in PCCs. We appreciate the valuable guidance provided by the reviewer.

1. Clinical, biochemistry data of 5 cases can be analysed. Any findings in different categories as per postulated classification can be noted for further studies. Example:epinephrine levels

We have now included the clinical information of 5 PCC patients, encompassing signs and symptoms, the tumor size, and laboratory test results in the revised manuscript as Supplemental Table S3 (Page 11-12). Notably, our analysis revealed that the kinase-type PCC patient (P4) exhibited higher blood pressures and plasma levels of catecholamine metabolites (3-methoxytyramine and normetanephrine) compared to metabolism-type PCC patients (P1-P3, and P5). This observation aligns with the elevated expression of phenylethanolamine N-methyltransferase (PNMT), an enzyme involved in the biosynthesis of catecholamine and linked to hypertension, in P4, as identified in the scRNA-seq data (Figure 4B and 4D) (Kennedy et al, 1993; Konosu-Fukaya et al, 2018; Nguyen et al, 2015). As suggested, we plan to conduct further research to explore the correlation of our molecular classification with plasma levels of catecholamine metabolites, and the relevant points have been discussed in the revision (Page 20).

We would like to take this chance to again thank the reviewer for the careful review and very helpful guidance about how to improve our study.

References for Reviewer #1:

Farrugia FA, Charalampopoulos A (2019) Pheochromocytoma. Endocrine regulations 53: 191-212Neumann HPH, Young WF, Jr., Eng C (2019) Pheochromocytoma and Paraganglioma. The New England journal of medicine 381: 552-565

Kennedy B, Elayan H, Ziegler MG (1993) Glucocorticoid hypertension and nonadrenal phenylethanolamine N-methyltransferase. Hypertension (Dallas, Tex : 1979) 21: 415419

Konosu-Fukaya S, Omata K, Tezuka Y, Ono Y, Aoyama Y, Satoh F, Fujishima F, Sasano H, Nakamura Y (2018) Catecholamine-Synthesizing Enzymes in Pheochromocytoma and Extraadrenal Paraganglioma. Endocrine pathology 29: 302309

Nguyen P, Khurana S, Peltsch H, Grandbois J, Eibl J, Crispo J, Ansell D, Tai TC (2015) Prenatal glucocorticoid exposure programs adrenal PNMT expression and adult hypertension. The Journal of endocrinology 227: 117-127

**Reviewer #2 (Recommendations for The Authors):**
1. Please revise all references to "malignant potential", "malignant behavior", etc. throughout the article, including the abstract and introduction, and replace them with the word "metastasis" as appropriate. Since all PCCs are malignant non-epithelial neuroendocrine neoplasms originating from the paraganglia, which are themselves malignant tumors, it is unacceptable to describe them as "malignant potential" or "malignant potential". Please review the 2022 WHO/IARC classification and description of pheochromocytoma/paraganglioma (reference: Mete O, Asa SL, Gill AJ, Kimura N, de Krijger RR, Tischler A. Overview of the 2022 WHO Classification of Paragangliomas and Pheochromocytomas. Endocr Pathol. 2022;33(1):90-114.doi:10.1007/s12022-022-09704-6).

As suggested, we have replaced all occurrences of “malignant potential” or “malignant behavior” with “metastasis” throughout the revised manuscript. We have also included a citation to the 2022 WHO/IARC classification for further clarity.

Similarly, it is not advisable to use the PASS score to predict "malignant" PCC; this type of scoring system evaluates the "metastasis risk" or the "metastasis potential" of PCC.

We appreciate the reviewer for this insight and have revised our statements accordingly.

Also, "MALIGNANT CHAFFIN CELLS" needs to be modified; in fact, it is the "tumor cell of PCC" that the authors are trying to express.

As suggested, we have amended the term “malignant chromaffin cells” to “PCC cells” in the revised manuscript (Page 9-10).

1. How does the PASS score specifically relate to intra-tumor heterogeneity as reflected by scRNA-seq? In fact, the PASS score evaluates the histological or pathological invasiveness of PCC, and different sections of the same tumor tissue may have different histological manifestations, which may affect the score; however, scRNA-seq analyzes the cellular composition of the tumor, which is not the same as the information reflected by the PASS score. Both represent different levels and dimensions of intra-tumor heterogeneity and should be analyzed together. Please specifically list, one by one, the proportion of each item score of the PASS system and cell type of scRNA-seq for each sample and the results of the comparisons with each other to better present the conclusions.

As suggested, we have included the proportion of each item score from the PASS system in the revised manuscript as Supplemental Table S2 (Page 8). Integrating this data with the cell type composition of each sample from Figure 2B, our analysis suggests that intra-tumor heterogeneity, as assessed by the PASS system, is more extensive compared to scRNA-seq. We concur with the reviewer’s judgement that scRNA-seq analysis and PASS score represent different levels and dimensions of intratumor heterogeneity, and we have adjusted our claim throughout the revised manuscript accordingly (Page 8, 9, and 19).

1. Where is the specific mutation site of the VHL gene in patient 5? Please advise.

The VHL gene mutation site, c.499C>T (missense mutation), in patient 5 was identified through whole exome sequencing (WES) analysis. We have now added the information to Supplemental Table S1 in the revised manuscript (Page 6).

1. Please revise Supplementary Figure 1, the scale should not appear in the picture of the staining result of P5.

As suggested, we have adjusted the position of the scale bar.

**Author response image 1. sa3fig1:** Hematoxylin-eosin staining and immunohistochemistry staining of CGA marker in formalin-fixed paraffin-embedded PCC tissue sections matched to scRNA-seq specimens. Scale bar, 100 μm.

1. What were the clinical presentation and biochemical findings in the five patients?

The information regarding tumor sizes, signs and symptoms, and plasma levels of catecholamine metabolites [3-methoxytyramine (3-MT), metanephrine (MN), and normetanephrine (NMN)] has been added to the revised manuscript as Supplemental Table S3 (Page 11-12).

Were there any preoperative symptoms of hypertension?

With the exception of P2, preoperative symptoms of hypertension were observed in all PCC patients. The information has been added to the revised manuscript as Supplemental Table S3 (Page 11-12).

What was the size and catecholamine secretion phenotype of each tumor? What was the relationship between these data and the scRNA-seq results?

The secretion phenotype showed that the kinase-type PCC patient (P4) exhibited higher plasma levels of catecholamine metabolites (3-methoxytyramine and normetanephrine) compared to metabolism-type PCC patients (P1-P3, and P5). This observation aligns with the elevated expression of phenylethanolamine Nmethyltransferase (PNMT), an enzyme involved in the biosynthesis of catecholamine and linked to hypertension, in P4, as identified in the scRNA-seq data (Figure 4B and 4D) (Kennedy et al, 1993; Konosu-Fukaya et al, 2018; Nguyen et al, 2015). Meanwhile, we have not observed the correlation between tumor sizes and molecular classification. We have now included tumor sizes and laboratory test results of 5 PCC patients in the revised manuscript as Supplemental Table S3 (Page 11-12), and the relevant points have been discussed in the revision (Page 20).

1. Please revise Figure 1A, the meaning shown in the figure appears to dissociate the tissues of the patient's normal adrenal glands, which can be misleading.

We appreciate the reviewer for raising this concern. The schematic in Figure 1A has been revised accordingly.

**Author response image 2. sa3fig2:** Schematic of the experimental pipeline. 11 tumor specimens and 5 adjacent normal adrenal medullary specimens were isolated from 5 PCC patients, dissociated into single-cell suspensions, and analyzed using 10x Genomics Chromium droplet scRNA-seq.

Please revise the figure note for Figure 1B, where the symbol (B) appears twice.

As suggested, we have revised the figure legends for Figure 1B and 1C (Page 42).

1. Please indicate in the figure legends and text what exactly is meant by "adjacent specimens"? medulla? cortex? normal tissue? I believe the authors mean adjacent normal adrenal medullary tissue, please check the article.

As suggested, we have revised the term “adjacent specimens” to “adjacent normal adrenal medullary tissues” throughout the revised manuscript.

1. Please review the pathologic diagnostic criteria of this study in light of the 2022 WHO/IARC guidelines for pathologic diagnosis: "For the pathological diagnosis, the inclusion criteria were neuroendocrine neoplasm originating from the adrenal medulla and retroperitoneal origin, i.e. pheochromocytoma and paraganglioma, with consistent morphologic and immunohistochemical confirmation in relevant cases and positivity for chromogranin A and synaptophysin. The exclusion criteria were adrenocortical neoplasm and metastatic tumors." It is not rigorous enough to diagnose a tumor as PCC based on positive CgA immunohistochemical staining results alone.

We have revised the statements about pathologic diagnostic criteria in accordance with the suggestion and have cited the reference (Page 6).

We would like to express our gratitude to the reviewer for the thorough review and invaluable guidance provided to enhance the quality of our study.

References for Reviewer #2:

Kennedy B, Elayan H, Ziegler MG (1993) Glucocorticoid hypertension and nonadrenal phenylethanolamine N-methyltransferase. Hypertension (Dallas, Tex: 1979) 21: 415419

Konosu-Fukaya S, Omata K, Tezuka Y, Ono Y, Aoyama Y, Satoh F, Fujishima F, Sasano H, Nakamura Y (2018) Catecholamine-Synthesizing Enzymes in Pheochromocytoma and Extraadrenal Paraganglioma. Endocrine pathology 29: 302309

Nguyen P, Khurana S, Peltsch H, Grandbois J, Eibl J, Crispo J, Ansell D, Tai TC (2015) Prenatal glucocorticoid exposure programs adrenal PNMT expression and adult hypertension. The Journal of endocrinology 227: 117-127

**Reviewer #3 (Recommendations For The Authors):**
I have several concerns and suggestions, which if addressed would improve the manuscript.1. The statements of “plasmas” in the manuscript and figures are confusing, which should be revised as “plasma cells”.

As suggested, we have revised the terminology from “plasmas” to “plasma cells” throughout the revised manuscript and figures.

1. The marker genes used for defining plasma cells (IGHG1 and IGLC2) showed low expressing percentage in Figure 1D. Please consider providing other genes as the marker of plasma cells.

As suggested, we performed additional analysis to pinpoint marker genes for accurate definition of plasma cells. Applying stricter statistical criteria (cut-off pvalue < 0.05, log2 fold change ≥ 1.5, and expressing percentage ≥ 0.6), we identified XBP1 (a transcription factor playing key roles in the final stages of plasma cell development) and IGKC (a type of light-chain immunoglobulins) (Todd et al, 2009; Poulsen et al, 2002) as top significant differentially expressed genes (DEGs) suitable for defining plasma cells. These data are now presented as Figure 1D in the revised manuscript (Page 7).

**Author response image 3. sa3fig3:** Dot plot of representative marker genes for each cell type. The color scale represents the average marker gene expression level; dot size represents the percentage of cells expressing a given marker gene.

1. The statement “Our clustering and cell type annotation analysis identified diverse adrenal cells, stromal cells, and immune cells within the PCC microenvironment” seems not be exhibited in Figure 1, so the clustering result of adrenal cells, stromal cells, and immune cells need to be added.

As suggested, we performed clustering analysis for adrenal cells, stromal cells, and immune cells (including lymphocytes and myeloid cells), and visualized by the Uniform Manifold Approximation and Projection (UMAP) plot. These data have been added to the revised manuscript as Supplemental Figure S3 (Page 8).

**Author response image 4. sa3fig4:** Integration Analysis across 5 PCC Patients Revealing the Cell Type Composition of the PCC Microenvironment. UMAP plot depicting the distribution of adrenal cells, stromal cells, and immune cells (including lymphocytes and myeloid cells) within the PCC microenvironment.

1. Given the classification of “metabolism-type PCCs” and “kinase-type PCCs” have not been presented in Figure 2D, the statement “Combined with our findings of a higher proportion of neutrophils and monocyts/macrophages in metabolism-type as compared with kinase-type” in Result 6 should be supported by using additional data.

As suggested, we performed additional analysis to evaluate the proportion of neutrophils and monocytes/macrophages in metabolism-type and kinasetype PCC patients. These data have been added to the revised manuscript as Supplemental Figure S4 (Page 14).

**Author response image 5. sa3fig5:** The frequency distribution of cell types within the microenvironment of metabolism-type and kinase-type PCC patients.

1. What makes the difference of scRNA-seq analysis and multispectral immunofluorescent staining in judging the immune escape of PCCs? Please provide an explanation.

We appreciate the reviewer's concern. scRNA-seq lacks spatial details, and multispectral immunofluorescent staining is constrained in the number of detected proteins. To address this, we employed both methods for analysis. scRNA-seq revealed limited communication between tumor and T cells, with lower HLA-I expression in kinase-type PCCs compared to metabolism-type PCCs. This was supported by multispectral staining using antibodies against CD4+ T cells, CD8+ T cells, M1 macrophages, or M2 macrophages markers, indicating sparse immune cell infiltration around tumor cells, mainly in the stroma (Figure 7A and 7B). This dual approach strengthens our understanding of immune escape in both PCC types. The explanation has been added to the revised manuscript (Page 21).

1. Figure 7G missed the scale bar for the staining results of marker proteins. Please add the scale bar into the figure.

As suggested, we have added to the scale bar accordingly.

1. In the method part of the manuscript, the authors should describe the minimum and maximum number used for quality control of the number of genes and the percentage of mitochondrial genes.

For quality control, we established a minimum threshold of no less than 200 genes and a maximum threshold of no more than 5000 genes. Additionally, the quality control process included a maximum threshold of 30% for mitochondrial genes. These specific criteria have been added to the methods section of the revised manuscript (Page 25-26).

We express our gratitude to the reviewer for their supportive recommendations and invaluable guidance on enhancing the rigor of our data.

References for Reviewer #3:

Todd DJ, McHeyzer-Williams LJ, Kowal C, Lee AH, Volpe BT, Diamond B, McHeyzer-Williams MG, Glimcher LH (2009) XBP1 governs late events in plasma cell differentiation and is not required for antigen-specific memory B cell development. The Journal of experimental medicine 206: 2151-2159

Poulsen TS, Silahtaroglu AN, Gisselø CG, Tommerup N, Johnsen HE (2002) Detection of illegitimate rearrangements within the immunoglobulin light chain loci in B cell malignancies using end sequenced probes. Leukemia 16: 2148-2155